# InfoBot: Transfer and Exploration via the Information Bottleneck

**Anirudh Goyal[1], Riashat Islam[2], Daniel Strouse[3], Zafarali Ahmed[2],**
**Matthew Botvinick[3, 5], Hugo Larochelle[1, 4], Yoshua Bengio[1], Sergey Levine[6]**

## ABSTRACT

A central challenge in reinforcement learning is discovering effective policies for tasks where rewards are sparsely distributed. We postulate that in the absence of useful reward signals, an effective exploration strategy should seek out *decision states*. These states lie at critical junctions in the state space from where the agent can transition to new, potentially unexplored regions. We propose to learn about decision states from prior experience. By training a goal-conditioned policy with an information bottleneck, we can identify decision states by examining where the model actually leverages the goal state. We find that this simple mechanism effectively identifies decision states, even in partially observed settings. In effect, the model learns the sensory cues that correlate with potential subgoals. In new environments, this model can then identify novel subgoals for further exploration, guiding the agent through a sequence of potential decision states and through new regions of the state space.

## 1 INTRODUCTION

Deep reinforcement learning has enjoyed many recent successes in domains where large amounts of training time and a dense reward function are provided. However, learning to quickly perform well in environments with sparse rewards remains a major challenge. Providing agents with useful signals to pursue in lieu of environmental reward becomes crucial in these scenarios. In this work, we propose to incentivize agents to learn about and exploit multi-goal task structure in order to efficiently explore in new environments. We do so by first training agents to develop useful habits as well as the knowledge of when to break them, and then using that knowledge to efficiently probe new environments for reward.

We focus on multi-goal environments and goal-conditioned policies (Foster and Dayan, 2002; Schaul et al., 2015; Plappert et al., 2018). In these scenarios, a goal $G$ is sampled from $p(G)$ and the beginning of each episode and provided to the agent. The goal $G$ provides the agent with information about the environment's reward structure for that episode. For example, in spatial navigation tasks, $G$ might be the location or direction of a rewarding state. We denote the agent's policy $\pi_\theta(A \mid S, G)$, where $S$ is the agent's state, $A$ the agent's action, and $\theta$ the policy parameters.

We incentivize agents to learn task structure by training policies that perform well under a variety of goals, while not overfitting to any individual goal. We achieve this by training agents that, in addition to maximizing reward, minimize the policy dependence on the individual goal, quantified by the conditional mutual information $I(A; G \mid S)$. This is inspired by the information bottleneck approach (Tishby et al., 1999) of training deep neural networks for supervised learning (Alemi et al., 2017; Chalk et al., 2016; Achille and Soatto, 2016; Kolchinsky et al., 2017), where classifiers are trained to achieve high accuracy while simultaneously encoding as little information about the input as possible. This form of "information dropout" has been shown to promote generalization performance (Achille and Soatto, 2016; Alemi et al., 2017). We show that minimizing goal information promotes generalization in an RL setting as well. Our proposed model is referred as InfoBot (inspired from the Information Bottleneck framework).

---

[1] Mila, University of Montreal,[2] Mila, McGill University, [3] Deepmind, [4] Google Brain, [5] University College London, [6] University of California, Berkeley. Corresponding author :`anirudhgoyal9119@gmail.com`

This approach to learning task structure can also be interpreted as encouraging agents to follow a *default policy*: This is the default behaviour which the agent should follow in the absence of any additional task information (like the goal location, the relative distance to the goal or a language instruction etc). To see this, note that our regularizer can also be written as $I(A; G \mid S) = \mathbb{E}_{\pi_\theta} [D_{\mathrm{KL}}[\pi_\theta(A \mid S, G) \mid \pi_0(A \mid S)]]$, where $\pi_\theta(A \mid S, G)$ is the agent's multi-goal policy, $\mathbb{E}_{\pi_\theta}$ denotes an expectation over trajectories generated by $\pi_\theta$, $D_{\mathrm{KL}}$ is the Kuhlback-Leibler divergence, and $\pi_0(A \mid S) = \sum_g p(g) \pi_\theta(A \mid S, g)$ is a "default" policy with the goal marginalized out. While the agent never actually follows the default policy $\pi_0$ directly, it can be viewed as what the agent might do in the absence of any knowledge about the goal. Thus, our approach encourages the agent to learn useful behaviours and to follow those behaviours closely, except where diverting from doing so leads to significantly higher reward. Humans too demonstrate an affinity for relying on default behaviour when they can (Kool and Botvinick, 2018), which we take as encouraging support for this line of work (Hassabis et al., 2017).

We refer to states where diversions from default behaviour occur as *decision states*, based on the intuition that they require the agent not to rely on their default policy (which is goal agnostic) but instead to make a goal-dependent "decision." Our approach to exploration then involves encouraging the agent to seek out these decision states in new environments. Decision states are natural subgoals for efficient exploration because they are boundaries between achieving different goals (van Dijk and Polani, 2011). By visiting decision states, an agent is encouraged to 1) follow default trajectories that work across many goals (i.e could be executed in multiple different contexts) and 2) uniformly explore across the many "branches" of decision-making. We encourage the visitation of decision states by first training an agent with an information regularizer to recognize decision states. We then freeze the agent's policy, and use $D_{\mathrm{KL}}[\pi_\theta(A \mid S, G) \mid \pi_0(A \mid S)]$ as an exploration bonus for training a *new* policy. Crucially, this approach to exploration is tuned to the family of tasks the agent is trained on, and we show that it promotes efficient exploration than other task-agnostic approaches to exploration (Houthooft et al., 2016; Pathak et al., 2017b).

Our contributions can be summarized as follows :

- We regularize RL agents in multi-goal settings with $I(A; G \mid S)$, an approach inspired by the information bottleneck and the cognitive science of decision making, and show that it promotes *generalization* across tasks.

- We use policies as trained above to then provide an *exploration bonus* for training new policies in the form of $D_{\mathrm{KL}}[\pi_\theta(A \mid S, G) \mid \pi_0(A \mid S)]$, which encourages the agent to seek out decision states. We demonstrate that this approach to exploration performs more effectively than other state-of-the-art methods, including a count-based bonus, VIME (Houthooft et al., 2016), and curiosity (Pathak et al., 2017b).

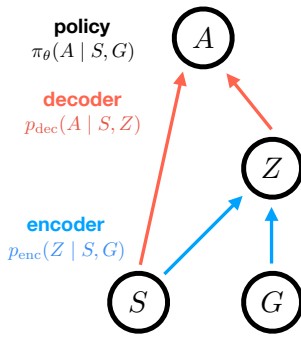

Figure 1: Policy architecture.

## 2 OUR APPROACH

Our objective is to train an agent on one set of tasks (environments) $T \sim p_{\mathrm{train}}(T)$, but to have the agent perform well on another different, but related, set of tasks $T \sim p_{\mathrm{test}}(T)$. We propose to maximize the following objective in the training environments:

$$
\begin{aligned}
J(\theta) &\equiv \mathbb{E}_{\pi_\theta}[r] - \beta I(A; G \mid S) \\
&= \mathbb{E}_{\pi_\theta}[r - \beta D_{\mathrm{KL}}[\pi_\theta(A \mid S, G) \mid \pi_0(A \mid S)]],
\end{aligned}
\tag{1}
$$

where $\mathbb{E}_{\pi_\theta}$ denotes an expectation over trajectories generated by the agent's policy, $\beta > 0$ is a tradeoff parameter, $D_{\mathrm{KL}}$ is the Kullback–Leibler divergence, and $\pi_0(A \mid S) \equiv \sum_g p(g) \pi_\theta(A \mid S, g)$ is a "default" policy with the agent's goal marginalized out.

## 2.1 Tractable bounds on information

We parameterize the policy $\pi_\theta(A \mid S, G)$ using an encoder $p_{\text{enc}}(Z \mid S, G)$ and a decoder $p_{\text{dec}}(A \mid S, Z)$ such that $\pi_\theta(A \mid S, G) = \sum_z p_{\text{enc}}(z \mid S, G) \, p_{\text{dec}}(A \mid S, z)$ (see figure 1). The encoder output $Z$ is meant to represent the information about the present goal $G$ that the agent believes is important to access in the present state $S$ in order to perform well. The decoder takes this encoded goal information and the current state and produces a distribution over actions $A$.

We suppress the dependence of $p_{\text{enc}}$ and $p_{\text{dec}}$ on $\theta$, but $\theta$ in the union of their parameters. Due to the data processing inequality (DPI) (Cover and Thomas, 2006), $I(Z; G \mid S) \geq I(A; G \mid S)$. Therefore, minimizing the goal information encoded by $p_{\text{enc}}$ also minimizes $I(A; G \mid S)$.

Thus, we instead maximize this lower bound on $J(\theta)$:

$$
\begin{aligned}
J(\theta) &\geq \mathbb{E}_{\pi_\theta}[r] - \beta I(Z; G \mid S) \\
&= \mathbb{E}_{\pi_\theta}[r - \beta D_{\text{KL}}[p_{\text{enc}}(Z \mid S, G) \mid p(Z \mid S)]],
\end{aligned} \tag{2}
$$

where $p(Z \mid S) = \sum_g p(g) \, p_{\text{enc}}(Z \mid S, g)$ is the marginalized encoding.

In practice, performing this marginalization over the goal may often be prohibitive, since the agent might not have access to the goal distribution $p(G)$, or even if the agent does, there might be many or a continuous distribution of goals that makes the marginalization intractable. To avoid this marginalization, we replace $p(Z \mid S)$ with a variational approximation $q(Z \mid S)$ (Kingma and Welling, 2014; Alemi et al., 2017; Houthooft et al., 2016; Strouse et al., 2018). This again provides a lower bound on $J(\theta)$ since:

$$
\begin{aligned}
I(Z; G \mid S) &= \sum_{z,s,g} p(z,s,g) \log \frac{p(z \mid g, s)}{p(z \mid s)} \\
&= \sum_{z,s,g} p(z,s,g) \log p(z \mid g, s) - \sum_{z,s} p(s) \, p(z \mid s) \log p(z \mid s) \\
&\geq \sum_{z,s,g} p(z,s,g) \log p(z \mid g, s) - \sum_{z,s} p(s) \, p(z \mid s) \log q(z \mid s),
\end{aligned} \tag{3}
$$

where the inequality in the last line, in which we replace $p(z \mid s)$ with $q(z \mid s)$, follows from that $D_{\text{KL}}[p(Z \mid s) \mid q(Z \mid s)] \geq 0 \Rightarrow \sum_z p(z \mid s) \log p(z \mid s) \geq \sum_z p(z \mid s) \log q(z \mid s)$.

Thus, we arrive at the lower bound $\tilde{J}(\theta)$ that we maximize in practice:

$$
J(\theta) \geq \tilde{J}(\theta) \equiv \mathbb{E}_{\pi_\theta}[r - \beta D_{\text{KL}}[p_{\text{enc}}(Z \mid S, G) \mid q(Z \mid S)]]. \tag{4}
$$

In the experiments below, we fix $q(Z \mid S)$ to be unit Gaussian, however it could also be learned, in which case its parameters should be included in $\theta$. Although our approach is compatible with any RL method, we maximize $\tilde{J}(\theta)$ on-policy from sampled trajectories using a score function estimator (Williams, 1992; Sutton et al., 1999a). As derived by Strouse et al. (2018), the resulting update at time step $t$, which we denote $\nabla_\theta \tilde{J}(t)$, is:

$$
\nabla_\theta \tilde{J}(t) = \tilde{R}_t \log(\pi_\theta(a_t \mid s_t, g_t)) - \beta \nabla_\theta D_{\text{KL}}[p_{\text{enc}}(Z \mid s_t, g_t) \mid q(Z \mid s_t)], \tag{5}
$$

where $\tilde{R}_t \equiv \sum_{u=t}^{T} \gamma^{u-t} \tilde{r}_u$ is a modified return, $\tilde{r}_t \equiv r_t + \beta D_{\text{KL}}[p_{\text{enc}}(Z \mid s_t, g) \mid q(Z \mid s_t)]$ is a modified reward, $T$ is the length of the current episode, and $a_t$, $s_t$, and $g_t$ are the action, state, and goal at time $t$, respectively. The first term in the gradient comes from applying the REINFORCE update to the modified reward, and can be thought of as encouraging the agent to change the policy in the present state to revisit future states to the extent that they provide high external reward as well as low need for encoding goal information. The second term comes from directly optimizing the policy to not rely on goal information, and can be thought of as encouraging the agent to directly alter the policy to avoid encoding goal information in the present state. Note that while we take a Monte Carlo policy gradient, or REINFORCE, approach here, our regularizer is compatible with any RL algorithm.

---

In practice, we estimate the marginalization over $Z$ using a single sample throughout our experiments.

## 2.2 POLICY AND EXPLORATION TRANSFER

By training the policy as in equation 5 the agent learns to rely on its (goal-independent) habits as much as possible, deviating only in decision states (as introduced in Section 1) where it makes goal-dependent modifications. We demonstrate in Section 4 that this regularization alone already leads to generalization benefits (that is, increased performance on $T \sim p_{\text{test}}(T)$ after training on $T \sim p_{\text{train}}(T)$). However, we train the agent to identify decision states as in equation 5, such that the learnt goal-dependent policy can provide an exploration bonus in the new environments. That is, after training on $T \sim p_{\text{train}}(T)$, we freeze the agent's encoder $p_{\text{enc}}(Z \mid S, G)$ and marginal encoding $q(Z \mid S)$, discard the decoder $p_{\text{dec}}(A \mid S, Z)$, and use the encoders to provide $D_{\text{KL}} \left[ p_{\text{enc}}(Z \mid S, G) \mid q(Z \mid S) \right]$ as a state and goal dependent exploration bonus for training a new policy $\pi_\phi(A \mid S, G)$ on $T \sim p_{\text{test}}(T)$. To ensure that the new agent does not pursue the exploration bonus solely (in lieu of reward), we also decay the bonus with continued visits by weighting with a count-based exploration bonus as well. That is, we divide the KL divergence by $\sqrt{c(S)}$, where $c(S)$ is the number of times that state has been visited during training, which is initialized to 1. Letting $r_e(t)$ be the environmental reward at time $t$, we thus train the agent to maximize the combined reward $r_t$:

$$ r_t = r_e(t) + \frac{\beta}{\sqrt{c(s_t)}} D_{\text{KL}} \left[ p_{\text{enc}}(Z \mid s_t, g_t) \mid q(Z \mid s_t) \right]. \tag{6} $$

Our approach is summarized in algorithm 1.

---

**Algorithm 1** Transfer and Exploration via the Information Bottleneck

---

**Require:**   A policy $\pi_\theta(A \mid S, G) = \sum_z p_{\text{enc}}(z \mid S, G) \, p_{\text{dec}}(A \mid S, z)$, parameterized by $\theta$
**Require:**   A variational approximation $q(Z \mid S)$ to the goal-marginalized encoder
**Require:**   A regularization weight $\beta$
**Require:**   Another policy $\pi_\phi(A \mid S, G)$, along with a RL algorithm $\mathcal{A}$ to train it
**Require:**   A set of training tasks (environments) $p_{\text{train}}(T)$ and test tasks $p_{\text{test}}(T)$
**Require:**   A goal sampling strategy $p(G \mid T)$ given a task $T$
   **for** episodes = 1 to $N_{\text{train}}$ **do**
      Sample a task $T \sim p_{\text{train}}(T)$ and goal $G \sim p(G \mid T)$
      Produce trajectory $\tau$ on task $T$ with goal $G$ using policy $\pi_\theta(A \mid S, G)$
      Update policy parameters $\theta$ over $\tau$ using Eqn 5
   **end for**
   *Optional*: use $\pi_\theta$ directly on tasks sampled from $p_{\text{test}}(T)$
   **for** episodes = 1 to $N_{\text{test}}$ **do**
      Sample a task $T \sim p_{\text{test}}(T)$ and goal $G \sim p(G \mid T)$
      Produce trajectory $\tau$ on task $T$ with goal $G$ using policy $\pi_\phi(A \mid S, G)$
      Update policy parameters $\phi$ using algorithm $\mathcal{A}$ to maximize the reward given by Eqn 6
   **end for**

---

## 3 RELATED WORK

van Dijk and Polani (2011) were the first to point out the connection between action-goal information and the structure of decision-making. They used information to identify decision states and use them as subgoals in an options framework (Sutton et al., 1999b). We build upon their approach by combining it with deep reinforcement learning to make it more scaleable, and also modify it by using it to provide an agent with an exploration bonus, rather than subgoals for options.

Our decision states are similar in spirit to the notion of "bottleneck states" used to define subgoals in hierarchical reinforcement learning. A bottleneck state is defined as one that lies on a wide variety of rewarding trajectories (McGovern and Barto, 2001; Stolle and Precup, 2002) or one that otherwise serves to separate regions of state space in a graph-theoretic sense (Menache et al., 2002; Şimşek et al., 2005; Kazemitabar and Beigy, 2009; Machado et al., 2017). The latter definition is purely based on environmental dynamics and does not incorporate reward structure, while both definitions can lead to an unnecessary proliferation of subgoals. To see this, consider a T-maze in which the agent starts at the bottom and two possible goals exist at either end of the top of the T. All states in this setup are bottleneck states, and hence the notion is trivial. However, only the junction where the lower and

upper line segments of the T meet are a decision state. Thus, we believe the notion of a decision state is a more parsimonious and accurate indication of good subgoals than is the above notions of a bottleneck state. The success of our approach against state-of-the-art exploration methods (Section 4) supports this claim.

We use the terminology of information bottleneck (IB) in this paper because we limit (or bottleneck) the amount of goal information used by our agent's policy during training. However, the correspondence is not exact: while both our method and IB limit information into the model, we maximize *rewards* while IB maximizes *information* about a target to be predicted. The latter is thus a supervised learning algorithm. If we instead focused on imitation learning and replaced $\mathbb{E}[r]$ with $I(A^*; A \mid S)$ in Eqn 1, then our problem would correspond exactly to a variational information bottleneck (Alemi et al., 2017) between the goal $G$ and correct action choice $A^*$ (conditioned on $S$).

Whye Teh et al. (2017) trained a policy with the same KL divergence term as in Eqn 1. But this term is used completely differently context: Whye Teh et al. (2017) use a regularizer on the KL-divergence between action distributions of different policies to improve distillation, does not have any notion of goals, and is not concerned with exploration or with learning exploration strategies and transferring them to new domain. We use the variational information bottleneck, which has a KL divergence penalty on the difference between the posterior latent variable distribution and the prior. We are not distilling multiple policies. Parallel to our work, Strouse et al. (2018) also used Eqn 1 as a training objective, however their purpose was not to show better generalization and transfer, but instead to promote the sharing and hiding of information in a multi-agent setting. In concurrent work (Galashov et al., 2019) proposed a way to learn default policy which helps to enforce an inductive bias, and helps in transfer across different but related tasks.

Popular approaches to exploration in RL are typically based on: 1) injecting noise into action selection (e.g. epsilon-greedy, (Osband et al., 2016)), 2) encouraging "curiosity" by encouraging prediction errors of or decreased uncertainty about environmental dynamics (Schmidhuber, 1991; Houthooft et al., 2016; Pathak et al., 2017b), or 3) count-based methods which incentivize seeking out rarely visited states (Strehl and Littman, 2008; Bellemare et al., 2016; Tang et al., 2016; Ostrovski et al., 2017). One limitation shared by all of these methods is that they have no way to leverage experience on previous tasks to improve exploration on new ones; that is, their methods of exploration are not tuned to the family of tasks the agent faces. Our transferrable exploration strategies approach in algorithm 1 however does exactly this. Another notable recent exception is Gupta et al. (2018), which took a meta-learning approach to transferable exploration strategies.

## 4 EXPERIMENTAL RESULTS

In this section, we demonstrate the following experimentally:

- The goal-conditioned policy with information bottleneck leads to much better policy transfer than standard RL training procedures (direct policy transfer).

- Using decision states as an exploration bonus leads to better performance than a variety of standard task-agnostic exploration methods (transferable exploration strategies).

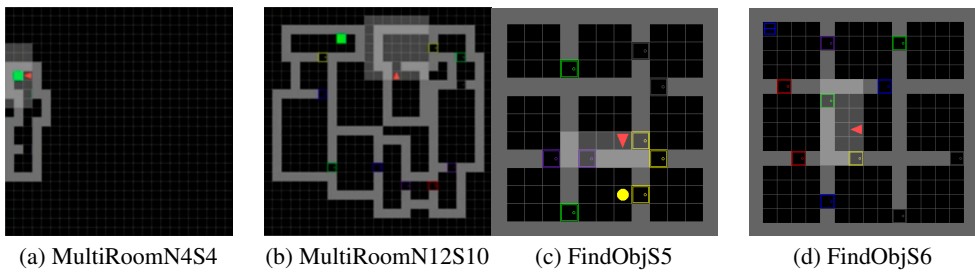

(a) MultiRoomN4S4          (b) MultiRoomN12S10          (c) FindObjS5          (d) FindObjS6

Figure 2: **MultiRoomN**$XS$**Y and FindObj**$SY$ **MiniGrid environments.** See text for details.

## 4.1 MiniGrid Environments

The first set of environments we consider are partially observable grid worlds generated with MiniGrid (Chevalier-Boisvert and Willems, 2018), an OpenAI Gym package (Brockman et al., 2016). We consider the **MultiRoomN$X$S$Y$** and a **FindObjS$Y$** task domains, as depicted in Figure 2. Both environments consist of a series of connected rooms, sometimes separated by doors that need opened. In both tasks, black squares are traversable, grey squares are walls, black squares with colored borders are doors, the red triangle is the agent, and the shaded area is the agent's visible region. The MultiRoomN$X$S$Y$ the environment consists of $X$ rooms, with size at most $Y$, connected in random orientations. The agent is placed in a distal room and must navigate to a green goal square in the most distant room from the agent. The agent receives an egocentric view of its surrounding, consisting of $3 \times 3$ pixels. The task increases in difficulty with $X$ and $Y$. The FindObjS$Y$ environment consists of 9 connected rooms of size $Y - 2 \times Y - 2$ arranged in a grid. The agent is placed in the center room and must navigate to an object in a randomly chosen outer room (e.g. yellow circle in bottom room in Figure 2c and blue square in top left room in Figure 2d). The agent again receives an egocentric observation, this time consisting of $7 \times 7$ pixels, and again the difficulty of the task increases with $Y$. For more details of the environment, see Appendix H.

Solving these partially observable, sparsely rewarded tasks by random exploration is difficult because there is a vanishing probability of reaching the goal randomly as the environments become larger. Transferring knowledge from simpler to more complex versions of these tasks thus becomes essential. In the next two sections, we demonstrate that our approach yields 1) policies that directly transfer well from smaller to larger environments, and 2) exploration strategies that outperform other task-agnostic exploration approaches.

## 4.2 Direct Policy Generalization on MiniGrid Tasks

We first demonstrate that training an agent with a goal bottleneck alone already leads to more effective policy transfer. We train policies on smaller versions of the MiniGrid environments (MultiRoomN2S6 and FindObjS5 and S7), but evaluate them on larger versions (MultiRoomN10S4, N10S10, and N12S10, and FindObjS7 and S10) throughout training.

Figure 3 compares an agent trained with a goal bottleneck (first half of Algorithm 1) to a vanilla goal-conditioned A2C agent (Mnih et al., 2016) on MultiRoomN$X$S$Y$ generalization. As is clear, the goal-bottlenecked agent generalizes much better. The success rate is the number of times the agent solves a larger task with 10-12 rooms while it is being trained on a task with only 2 rooms. When generalizing to 10 small rooms, the agent learns to solve the task to near perfection, whereas the goal-conditioned A2C baseline only solves <50% of mazes (Figure 3a).

Table 1 compares the same two agents on FindObjS$Y$ generalization. In addition, this comparison includes an ablated version of our agent with $\beta = 0$, that is an agent with the same architecture as in Figure 1 but with the no information term in its training objective. This is to ensure that our method's success is not due to the architecture alone. As is evident, the goal-bottlenecked agent again generalizes much better.

We analyzed the agent's behaviour to understand the intuition of why it generalizes well. In the MultiRoomN$X$S$Y$ environments, we find that the agent quickly discovers a wall following strategy. Since these environments are partially observable, this is indeed a good strategy that also generalizes well to larger mazes. In the FindObjS$Y$ environments, on the other hand, the agent sticks toward the center of the rooms, making a beeline from doorway to doorway. This is again a good strategy, because the agent's field of view in these experiments is large enough to see the entire room in which its in to determine if the goal object is present or not.

| Method | FindObjS7 | FindObjS10 |
|--------|-----------|------------|
| Goal-conditioned A2C | 56% | 36% |
| InfoBot with $\beta = 0$ | 44% | 24% |
| InfoBot | **81%** | **61%** |

Table 1: **Policy generalization on FindObjS$Y$.** Agents trained on FindObjS5, and evaluated on FindObjS7 and S10.

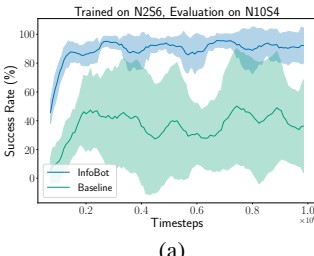 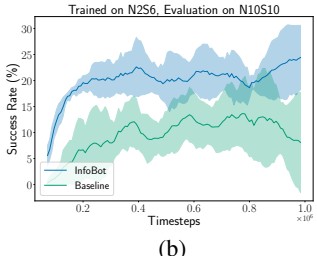 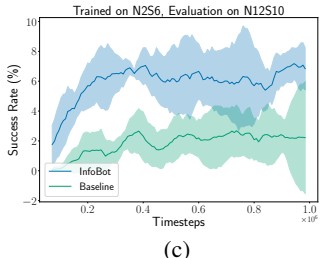

(a)         (b)         (c)

Figure 3: **Policy generalization on MultiRoomN$X$S$Y$.** Success is measured by the percent of time the agent can find the goal in an unseen maze. Error bars are standard deviations across runs. Baseline is a vanilla goal-conditioned A2C agent.

| Method | MultiRoomN3S4 | MultiRoomN5S4 |
|---|---|---|
| Goal-conditioned A2C | 0% | 0% |
| TRPO + VIME | 54% | 0% |
| Count based exploration | **95%** | 0% |
| Curiosity-based exploration | **95%** | 54% |
| InfoBot (decision state exploration bonus) | 90% | **85%** |

Table 2: **Transferable exploration strategies on MultiRoomN$X$S$Y$.** InfoBot encoder trained on MultiRoomN2S6. All agents evaluated on MultiRoomN3S4 and N5S4. While several methods perform well with 3 rooms, InfoBot performs far better as the number of rooms increases to 5.

## 4.3 Transferable Exploration Strategies on MiniGrid Tasks

We now evaluate our approach to exploration (the second half of Algorithm 1). We train agents with a goal bottleneck on one set of environments (MultiRoomN2S6) where they learn the sensory cues that correspond to decision states. We then use the identified decision states to guide exploration on another set of environments (MultiRoomN3S4, MultiRoomN4S4, and MultiRoomN5S4). We compare to several standard task-agnostic exploration methods, including count-based exploration (Eqn 6 without the $D_{\text{KL}}$, that a bonus of $\beta/\sqrt{c(s)}$), VIME (Houthooft et al., 2016), and curiosity-driven exploration (Pathak et al., 2017b), as well as a goal-conditioned A2C baseline with no exploration bonuses. Results are shown in Table 2 and Figure 4.

On a maze with three rooms, the count-based method and curiosity-driven exploration slightly outperform the proposed learned exploration strategy. However, as the number of rooms increases, the count-based method and VIME fail completely and the curiosity-based method degrades to only 54% success rate. This is in contrast to the proposed exploration strategy, which by learning the structure of the task, maintains a high success rate of 85%.

## 4.4 Goal-based Navigation Tasks

In this task, we use a partially observed goal based MiniPacMan environment as shown in Figure 5. The agent navigates a maze, and tries to reach the goal. The agent sees only a partial window around itself. The agent only gets a reward of "1" when it reaches the goal. For standard RL algorithms, these tasks are difficult to solve due to the partial observability of the environment, sparse reward (as the agent receives a reward only after reaching the goal), and low probability of reaching the goal via random walks (precisely because these junction states are crucial states where the right action must be taken and several junctions need to be crossed). This environment is more challenging as compared to the Minigrid environment, as this environment also has dead ends as well as more complex branching.

We first demonstrate that training an agent with a goal bottleneck alone leads to more effective policy transfer. We train policies on smaller versions of this goal based MiniPacMan environment environments (6 x 6 maze), but evaluate them on larger versions (11 X 11) throughout training.

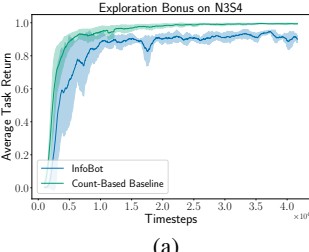 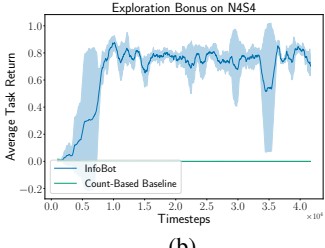 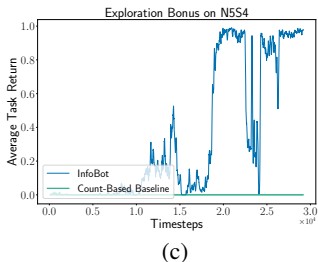

(a)             (b)             (c)

Figure 4: **Transferable exploration strategies on MultiRoomN$XS Y$.** As the number of rooms increases (from left to right), a count-based exploration bonus alone cannot solve the task, whereas the proposed exploration bonus, by being tuned to task structure, enables success on these more difficult tasks.

| Algorithm (Train on $6 \times 6$ maze) | Evaluate on $11 \times 11$ maze |
| --- | --- |
| Actor-Critic | 5% |
| PPO (Proximal Policy Optimization) | 8% |
| Actor-Critic + Count-Based | 7% |
| Curiosity Driven Learning (ICM) | 47% |
| Goal Based (UVFA) Goal - TopDownImage of the goal | 7% |
| Goal Based (UVFA) Goal - Relative Dist | 15% |
| Feudal RL | 37% |
| InfoBot (proposed) | 64% |

Table 3: Experiments for training the agent in a $6 \times 6$ maze environment, and then generalizing to a $11 \times 11$ maze. Comparison of our proposed method to regular actor-critic methods, UVFA and other hierarchical approaches. Results shown for the % of times agent reaches the goal. The results are average over 3 random seeds.

Table. 3 compares an agent trained with a goal bottleneck (first half of Algorithm 1) to a vanilla goal-conditioned A2C agent (Mnih et al., 2016), exploration methods like count-based exploration, curiosity driven exploration (Pathak et al., 2017a) and Feudal RL algorithm (Vezhnevets et al., 2017). The goal-bottlenecked agent generalizes much better. The success rate is the number of times the agent solves a larger task while it is being trained on a task with only 2 rooms. When generalizing to larger maze, the agent learns to solve the task 64% of the times, whereas other agents solve <50% of mazes (Table 3). For representing the goals, we experiment with 2 versions. 1) In which we give the agent's relative distance to the goal as the goal representation. 2) In which we give the top down image of the goal. For our experiments, the baseline in which we give the relative distance to the goal worked better.

## 4.5 MIDDLE GROUND BETWEEN MODEL BASED RL AND MODEL FREE RL

We further demonstrate the idea of decision states in a planning goal-based navigation task that uses a combination of model-based and model-free RL. Identifying useful decision states can provide a comfortable middle ground between model-free reasoning and model-based planning. For example, imagine planning over individual decision states, while using model-free knowledge to navigate between bottlenecks: aspects of the environment that are physically complex but vary little between problem instances are handled in a model-free way (the navigation between decision points), while the particular decision points that are relevant to the task can be handled by explicitly reasoning about causality, in a more abstract representation. We demonstrate this using a similar setup as in imagination augmented agents (Weber et al., 2017). In imagination augmented agents, model free agents are augmented with imagination, such that the imagination model can be queried to make predictions about the future. We use the dynamics models to simulate imagined trajectories, which are then summarized by a neural network and this summary is provided as additional context to a policy network. We use the output of the imagination module as a "goal" and we want to show that

only near the decision points (i.e potential subgoals) the agent wants to make use of the information which is a result of running imagination module.

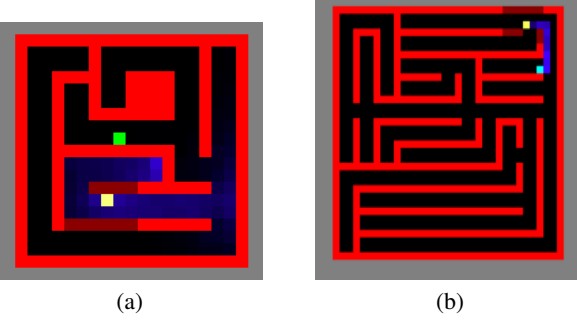

(a)                                                                        (b)

Figure 5: Goal based MiniPacMan navigation task: We train on a $6 \times 6$ environment, and evaluate the generalization performance in a $11 \times 11$ maze. The agent is represented by white color and has to reach the goal (light green marker).

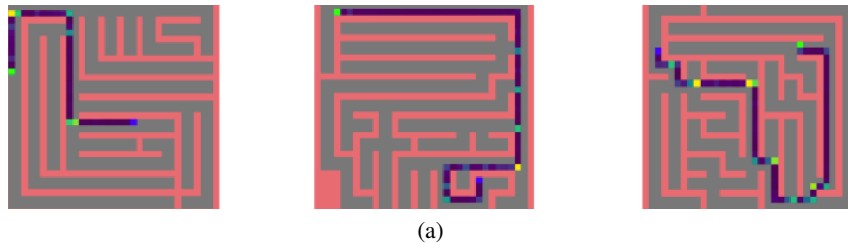

(a)

Figure 6: Goal based MiniPacMan navigation task: Here the agent gets a full observation of environment. We follow the similar setup as in Imagination Augmented agents. In this, the output of the imagination core is treated as a contextual information by the policy. We treat this contextual information as the "goal" in the InfoBot setup. Here, we want to see, where the policy wants to access the information provided by running the imagination module. Ideally, only at the decision states (i.e potential sub-goals) policy should access the output of the imagination module. We show the output of $D_{\text{KL}} \left[ p_{\text{enc}}(Z \mid s_t, g_t) \mid q(Z \mid s_t) \right]$, where $g_t$ refers to the output of imagination module. High KL is represented by lighter color.

## 5  CONNECTIONS TO NEUROSCIENCE AND COGNITIVE SCIENCE

The work we have presented bears some interesting connections to cognitive science and neuroscience. Both of these fields draw a fundamental distinction between automatic and controlled action selection (Miller and Cohen, 2001). In automatic responses, perceptual inputs directly trigger actions according to a set of habitual stimulus-response associations. In controlled behaviour, automatic responses are overridden in order to align behaviour with a more complete representation of task context, including current goals. As an example, on the drive to work, automatic responding may trigger the appropriate turn at a familiar intersection, but top-down control may be needed to override this response if the same intersection is encountered on the route to a less routine destination.

As is readily evident, our InfoBot architecture contains two pathways that correspond rather directly to the automatic and controlled pathways that have been posited in cognitive neuroscience models (Miller and Cohen, 2001). In the neuroscience context, representation of task context and the function of overriding automatic responses has been widely linked with the prefrontal cortex (Miller and Cohen, 2001), and it is interesting to consider the route within InfoBot from goal to action representations in this light. Notably, recent work has suggested that prefrontal control processes are associated with subjective costs; ceteris paribus, human decision-makers will opt for habitual or automatic routes to behaviour. This of course aligns with InfoBot, and in particular with the KL term in Equation 1.

This correspondence with neuroscience provides some indirect encouragement for the approach implemented in the present work. In turn, the InfoBot framework provides an indication for why a cost of control may exist in human cognition, namely that this encourages the emergence of useful habits, with payoffs for efficient exploration and transfer.

## 6    CONCLUSION

In this paper, we proposed to train agents to develop "default behaviours" as well as the knowledge of when to break those behaviour, using an information bottleneck between the agent's goal and policy. We demonstrated empirically that this training procedure leads to better direct policy transfer across tasks. We also demonstrated that the states in which the agent learns to deviate from its habits, which we call "decision states", can be used as the basis for a learned exploration bonus that leads to more effective training than other task-agnostic exploration methods.

### ACKNOWLEDGEMENTS

The authors acknowledge the important role played by their colleagues at Mila throughout the duration of this work. AG would like to thank Doina Precup, Maxime Chevalier-Boisvert and Alexander Neitz for very useful discussions. AG would like to thank Konrad Kording, Rosemary Nan Ke, Jonathan Binas, Bernhard Schoelkopf for useful discussions. The authors would also like to thank Yee Whye Teh, Peter Henderson, Emmanuel Bengio, Philip Bachman, Michael Noukhovitch for feedback on the draft. The authors are grateful to NSERC, CIFAR, Google, Samsung, Nuance, IBM, Canada Research Chairs, Canada Graduate Scholarship Program, Nvidia for funding, and Compute Canada for computing resources. We are very grateful to Google for giving Google Cloud credits used in this project.

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

## A MATHEMATICAL FRAMEWORK

We show that the proposed approach is equivalent to regularizing agents with a variational upper bound on the mutual information between goals and actions given states.

Recall that the variational information bottleneck objective (Alemi et al., 2016; Tishby et al., 2000) is formulated as the maximization of $I(Z, Y) - \beta I(Z, X)$. In our setting, the input $(X)$ corresponds to the goal of the agent $(G)$ and $(A)$ corresponds to the target output.

We assume that the joint distribution $p(G, A, Z|S)$ factorizes as follows: $p(G, A, Z|S) = p(Z|G, A, S)p(A|Z, S)p(G|S) = p(Z|G, S)p(A|Z, S)p(G|S)$ i.e., we assume $p(Z|G, A, S) = p(Z|G, S)$, corresponding to the Markov chain $G \rightarrow Z \rightarrow A$.

The Data Processing Inequality (DPI) (Beaudry and Renner, 2012; Kinney and Atwal, 2014; Achille and Soatto, 2017) for a Markov chain $x \rightarrow z \rightarrow y$ ensures that $I(x; z) \geq I(x; y)$.

Hence for Infobot, it implies,

$$I(A; G|S) \leq I(Z; G|S) \tag{7}$$

To get an upper bound on $I(G; Z|S)$, we first would get an upper bound on $I(G; Z|S = s)$, and then we average over $p(s)$ to get the required upper bound.

We get the following result,

$$I(G; Z|S = s) = \sum_{z,g} p(g|s)p(z|s, g) \log \frac{p(z|s, g)}{p(z|s)}, \tag{8}$$

Here, we assume that marginalizing over goals to get $p(z|s) = \sum_g p(g)p(z|s, g)$ is intractable, and so we approximate it with a normal prior $p_{prior} = N(0, 1)$. Since the cross-entropy between $p(z|s)$ and $p_{prior}(z)$ is larger than between $p(z|s)$ and itself, we get the following upper bound:

$$
\begin{aligned}
I(G; Z|S = s) &\leq \sum_g p(g|s) \sum_z p(z|s, g) \log \frac{p(z|s, g)}{p_{prior}(z)} \\
&= \sum_g p(g|s) D_{KL}[p(z|s, g)|p_{prior}(z)]
\end{aligned}
\tag{9}
$$

Averaging over state probabilities gives

$$
\begin{aligned}
I(Z; G|S) &\leq \sum_s p(s) \sum_g p(g|s) D_{KL}[p(z|s, g)|p_{prior}(z)] \\
&= \sum_g p(g) \sum_s p(s|g) D_{KL}[p(z|s, g)|p_{prior}(z)]
\end{aligned}
\tag{10}
$$

Using Eq. 7 , we can get an upper bound on the mutual information between immediate action and goal. Here $r(z)$ is a fixed prior.

$$
\begin{aligned}
I(A; G|S) &\leq I(Z; G|S) \\
&\leq \underbrace{\sum_g p(g)}_{\text{sample a goal}} \underbrace{\sum_s p(s|g)}_{\text{sample a trajectory}} \underbrace{\text{KL}[p_{\text{enc}}(z|s, g)||p_{prior}(z)]}_{\text{penalize encoder for departure from prior}} .
\end{aligned}
\tag{11}
$$

Hence, minimizing the KL between $p_{prior}(z)$ and $p(z|s, g)$ penalizes the use of information about the goal by the policy, so that when the policy decides to use information about the goal, it must be worthwhile, otherwise the agent is supposed to follow a "default" behaviour.

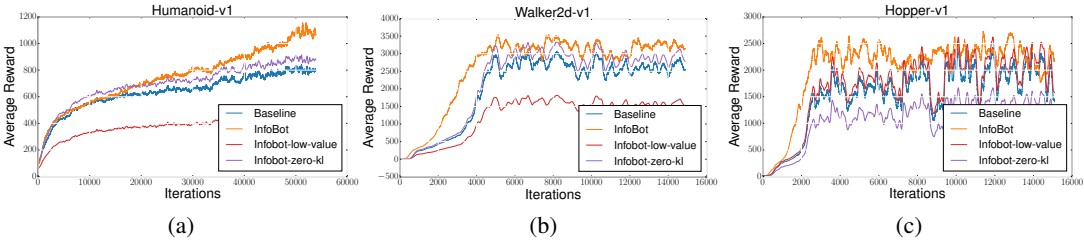

Figure 7: **Transferable exploration strategies on Humanoid, Walker2D, and Hopper.** The "baseline" is PPO (Schulman et al., 2017). Experiments are run with 5 random seeds and averaged over the runs.

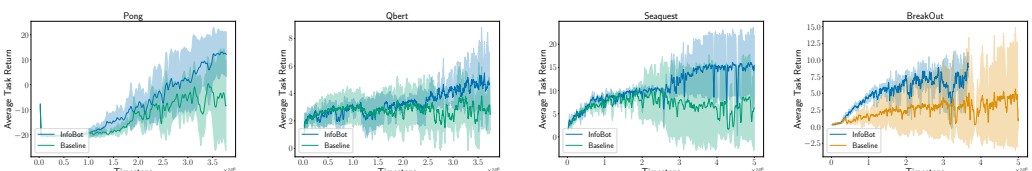

Figure 8: **Transferable exploration strategies on Pong, Qbert, Seaquest, and Breakout.** The baseline is a vanilla A2C agent. Results averaged over three random seeds.

### A.1 TRANSFERABLE EXPLORATION STRATEGIES FOR CONTINUOUS CONTROL

To show that the InfoBot architecture can also be applied to continuous control, we evaluated the performance of InfoBot on three continuous control tasks from OpenAI Gym (Brockman et al., 2016). Because InfoBot depends on the goal, in the control domains, we use high value states as an approximation to the goal state following Goyal et al. (2018). We maintain a buffer of high value states, and at each update, we sample a high value state from the buffer which acts as a proxy for the goal. We compared to proximal policy optimization (PPO) (Schulman et al., 2017), as well as two ablated versions of our model: 1) instead of taking high value states, we take low value states from the buffer as proxy to the goal ("InfoBot-low-value") and 2) we use the same InfoBot policy architecture but do not use the information regularizer (i.e. $\beta = 0$) ("InfoBot-zero-kl"). The results in Figure 7 show that InfoBot improves over all three alternatives.

### A.2 TRANSFERABLE EXPLORATION STRATEGIES FOR ATARI

We further evaluate our experiments on few Atari games (Bellemare et al., 2013) using A2C and compare it with our proposed InfoBot framework. In this experiment, our goal is to show that InfoBot can generalize to even more complex domains compared to the maze tasks above. As in the control experiments, we again use high value states as a proxy to the goal Goyal et al. (2018) and maintain a buffer of 20000 states to sample and prioritize high value states. We evaluate our proposed model on 4 Atari games (Pong, QBert, Seaquest and BreakOut) as shown in figure 8. We find that compared to a vanilla A2C agent, our InfoBot A2C agent performs significantly better on these Atari tasks. Our experiments are averaged over three random seeds.

## B TRANSFERRED EXPLORATION STRATEGY IN ATARI

We demonstrate that the encoder can be used to transfera exploration strategy across Atari games to help agents learn a new game quickly. To initialize the encoder, we train an agent on the game Seaquest, where the agent is trained to identify the decision states. We then re-use this encoder on another Atari environment to provide an exploration bonus. Our experiments on the Atari games are evaluated on the games of Pong and Qbert. On Pong, an agent with learns to get a task return of 20 in 3M time steps. We show that the identification of bottleneck states can be used to transfer exploration strategies across Atari games (Fig. 9).

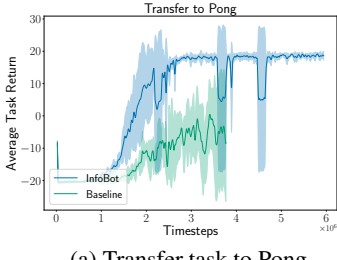
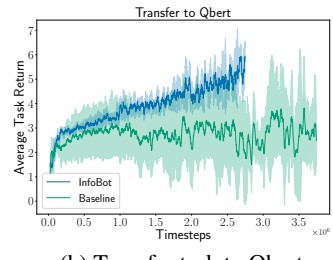

(a) Transfer task to Pong                     (b) Transfer task to Qbert

Figure 9: **Transfer across ALE Games** (Pong, Qbert and Freeway) using egocentric encoder to provide exploration bonus, trained from Seaquest. Comparison of InfoBot (A2C + KL Regularizer) with a Baseline A2C. Experiment results averaged over four random seeds. See Section B.

## C  ALGORITHM IMPLEMENTATION DETAILS

We evaluate the InfoBot framework using Adavantage Actor-Critic (A2C) to learn a policy $\pi_\theta(a|s,g)$ conditioned on the goal. To evaluate the performance of InfoBot, we use a range of maze multi-room tasks from the gym-minigrid framework (Chevalier-Boisvert and Willems, 2018) and the A2C implementation from (Chevalier-Boisvert and Willems, 2018). For the maze tasks, we used agent's relative distance to the absolute goal position as "goal".

For the maze environments, we use A2C with 48 parallel workers. Our actor network and critic networks consist of two and three fully connected layers respectively, each of which have 128 hidden units. The encoder network is also parameterized as a neural network, which consists of 1 fully connected layer. We use RMSProp with an initial learning rate of 0.0007 to train the models, for both InfoBot and the baseline for a fair comparison. Due to the partially observable nature of the environment, we further use a LSTM to encode the state and summarize the past observations.

For the Atari experiments, we use the open-source A2C implementation from Kostrikov (2018) and condition the goal state into the policy network. For the actor network, we use 3 convolution layers with ReLU activations. For training the networks, we use RMSProp with pre-defined learning rates for our algorithm and the baseline. The goal state in our experiments is used as the high value state following (Goyal et al., 2018).

For the Mujoco Experiments, we use Proximal Policy Optimization (PPO) (Schulman et al., 2017) with the open-source implementation available in Kostrikov (2018), using the defined architectures and hyperparameters as given in the repository for both our algorithm and the baseline. Again, for the goal-conditioned policies, the goal state for Mujoco experiments is defined as the high value state following (Goyal et al., 2018).

For reproducibility purposes of our experiments, we will further release the code on github that will be available on

**Hyperparameter Search for Grid World Experiments**: For the proposed method, we only varied the weight of KL loss. We tried 5 different values 0.1, 0.9, 0.01, 0.09, 0.005 for each of the env. and plotted the results which gave the most improvement. We used CNN policy for the case of FindObjSY env, and MLP policy for the case of MultiRoomNXSY. We used the agent's relative distance to goal as a goal for our goal conditioned policy.

**Hyperparameter Search for Control experiments** We have not done any hyper-parameter search for the baseline. For the proposed method, we only varied the weight of KL loss. We tried 3 different values 0.5, 0.01, 0.09 for each of the env. and plotted the results which gave the most improvement over the PPO baseline.

---

https://github.com/anonymous

| Method | Train Reward | TestReward |
|---|---|---|
| No Communication | -0.919 | -0.920 |
| Communication | -0.36 | -0.472 |
| Communication (with KL cost) | -0.293 | -0.38 |

Table 4: Training and test physical reward for setting with comunication, without communication, with limited communication (using InfoBot cost)

.

## D  MULTIAGENT COMMUNICATION

Here, we want to show that by training agents to develop "default behaviours" as well as the knowledge of when to break those behaviours, using an information bottleneck can also help in other scenarios like multi-agent communication. Consider multiagent communication, where in order to solve a task, agents require communicating with another agents. Ideally, an agent would would like to communicate with other agent, only when its essential to communicate, i.e the agents would like to minimize the communication with another agents. Here we show that selectively deciding when to communicate with another agent can result in faster learning.

In order to be more concrete, suppose there are two agents, Alex and Bob, and that Alex receives Bob's action at time t, then Alex can use Bob's action to decide what influence Bob's action has on its own action distribution. Lets say the current state of the Alex and Bob are represented by $s_a, s_r$ respectively, and the communication channel b/w Alex and Bob is represented by $z$. Alex and BOb can decide what action to take based on the distribution $p_a(a_a|s_s), p_r(a_r|s_r)$ respectively. Now, when Alex wants to use the information regarding Bob's action, then the modified action distribution (for Alex) becomes $p_a(a_a|s_a, z_r)$ where $z_r$ contains information regarding Bob's past states and actions, similarly the modified action distribution (for Bob) becomes $p_r(a_r|s_r, z_a)$. Now, the idea is that Alex and Bob wants to know about each other actions *only* when its necessary (i.e the goal is to minimize the communication b/w Alex and Bob) such that on average they only use information corresponding to there own states (which could include past states and past actions.) This would correspond to penalizing the difference between the marginal policy of Alex ("habit policy") and the conditional policy of Alex (conditioned on Bob's state and action information) as it tells us how much influence Bob's action has on Alex's action distribution. In mathematical terms, it would correspond to penalizing $D_{\mathrm{KL}}\left[p_a(a_a \mid s_a, z_r) \mid p_a(a_a \mid s_a)\right]$.

In order to show this, we use the same setup as in the paper (Mordatch and Abbeel, 2018). The agent perform actions and communicate utterances according to policy which is identically instantiated for all the different agents. This policy determines the action, and the communication protocols. We assume all agents have identical action and observation spaces, and all agents act according to the same policy and receive a shared reward. We consider the cooperative setting, in which the problem is to find a policy that maximizes expected return for all the agents.Table 4 shows the training and test rewards as compared to the scenarios when there is no communication b/w different agents, when all the agents can communication with each other, and when there is a KL cost penalizing the KL b/w the conditional policy distribution and marginal policy distribution. As evident, agents trained with InfoBot cost achieves the best results.

### D.1  MULTIAGENT COMMUNICATION

The environment consists of N agents and M landmarks. Both the agents and landmarks exhibit different characteristics such as different color and shape type. Different agents can act to move in the environment. They can also act be effected by the interactions with other agents. Asides from taking physical actions, agents communicate with other agents using verbal communication symbols $c$. We use the same setup as in the paper (Mordatch and Abbeel, 2018).

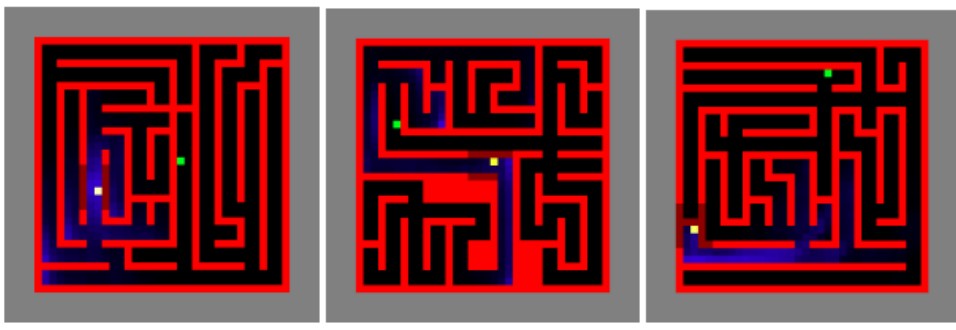

(a) InfoBot Visitation Count

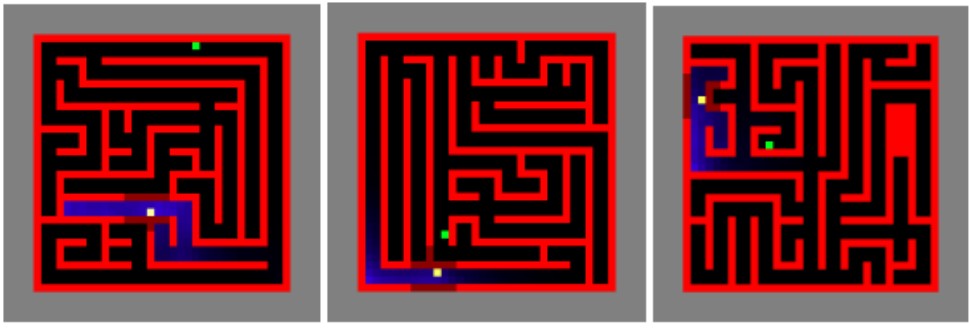

(b) Baseline Policy Visitation Count

Figure 10: Visitation Count: Effect on visitation count as a result of giving KL as an exploration bonus. As top figure shows, that after giving KL as an exploration bouns, agent visits more diverse states. Here, the agent is trained on a smaller 6 x 6 maze, and evaluated on more complex 11 * 11 maze. The "blueness" quanitifies the states visited by the agent.

## E    EFFECT ON VISITATION COUNT DUE TO EXPLORATION BONUS

## F    COMPARISON WITH OFF POLICY ALGORITHMS (SAC)

In this section, we study in isolation the effect of proposed method as compared to the state of the art off-policy methods like SAC (Haarnoja et al., 2018). For this domain, we again use high value states as an approximation to the goal state following (Goyal et al., 2018). In order to implement this, we maintain a buffer of 20000 states, and choose the state with the highest value under the current value function, as a proxy to the goal.

We compare the proposed method with SAC in sparse reward scenarios. We evaluate the proposed algorithm on 4 MuJoCo tasks in OpenAI Gym. (Brockman et al., 2016)

The result in Figure 11 shows the performance of the proposed method showing improvement over the baseline on HalfCheetah-v2, Walker2d-v2, Hopper-v2 and Swimmer-v2 sparse reward tasks. We evaluate the performance by evaluating after every 50K steps. We plotted the performance of the proposed method as well as baseline for 500K steps. We averaged over 2 random seeds.

## G    INFORMATION REGULARIZATION FOR INSTRUCTION FOLLOWING

Here we use the proposed method in the context of interactive worlds for spatial reasoning where the goal is given by language instruction. The agent is placed in an interactive world, and agent can take actions to reach the goal specified by language instruction. For ex. Reach the north-most house, the problem could be challenging because the language instruction is is highly context-

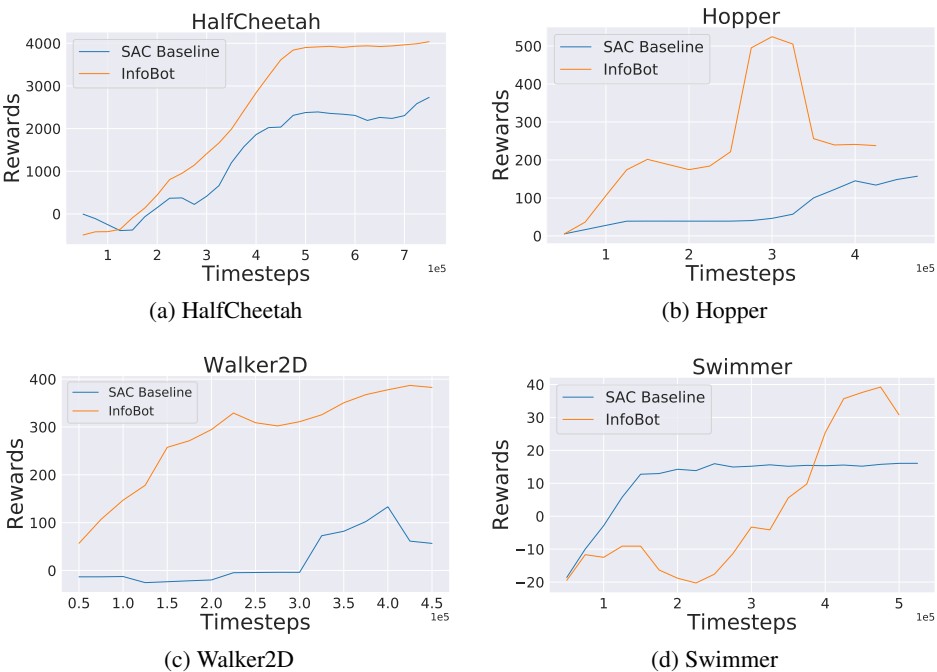

(a) HalfCheetah

(b) Hopper

(c) Walker2D

(d) Swimmer

Figure 11: InfoBot comparison with state of the art off policy algorithm (SAC) on sparse reward mujoco envs.

dependent. Therefore, for better generalization to unseen worlds, the model must jointly reason over the instruction text and environment configuration.

We model our task as a Markov Decision Process (MDP), where an agent can take actions to effect the world. The goal to the agent is specified by the language instruction. The MDP can be represented by the tuple (S,A,G,T,R), where S is the set of all possible states, A is the action set, G is the set of all goal specifications in natural language, $T(s_{next}|s, a, g)$ is the transition distribution, and R(s,g) is the reward function, which is dependent on both the current state as well as goal. A state s ∈ S includes information such as the locations of different entities along with the agent's own position.

**Puddle world navigation data** In order to study generalization across a wide variety of environmental conditions and linguistic inputs, we follow the same experimental setup as in (Janner et al., 2018). Its basically an extension of the puddle world reinforcement learning benchmark. States in a grid are first filled with either grass or water cells, such that the grass forms one connected component. We then populate the grass region with six unique objects which appear only once per map (triangle, star, diamond, circle, heart, and spade) and four non-unique objects (rock, tree, horse, and house) which can appear any num- ber of times on a given map.

We compare the proposed method to the UVFA (Schaul et al., 2015). We made use of one MLPs and the LSTM to learn low dimensional embeddings of states and goals respectively which are then combined via dot product to give value estimates. Goals are described in text described in text, and hence we use the LSTM over the language instruction. The state MLP has an identical architecture to that of the UVFA: two hidden layers of dimension 128 and ReLU activations.

Text instructions can have both local and global references to objects. Local references require an understanding of spatial prepositional phrases such as 'above', 'next to' in order to reach the goal. This is invariant to the global position of the object references, on the other hand, global references contains superlatives such as 'easternmost' and 'topmost', which require reasoning over the entire map. For example, in the case of local reference, one could describe a unique object (e.g. Go to the circle), whereas for global reference might require comparing the positions of all objects of a specific type (e.g. Go to the northernmost tree). Fig. 12 shows randomly generated worlds.

---

Image taken from https://github.com/JannerM/spatial-reasoning

| Method | Local - Policy Quality | Global - Policy Quality |
|---|---|---|
| UVFA | 0.57 | 0.59 |
| InfoBot (Proposed Method) | 0.89 | 0.81 |

Table 5: Performance of models trained via reinforcement learning on a held-out set of environments and instructions. Policy quality is the true expected normalized reward. We show results from training on the local and global instructions both separately and jointly.

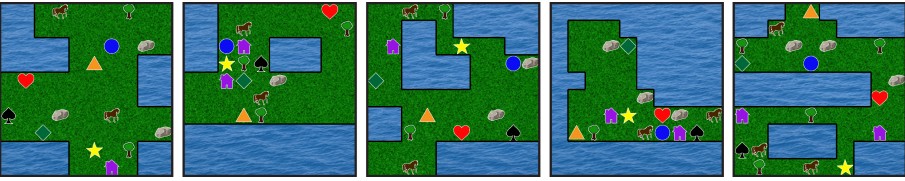

Figure 12: Visualizations of randomly generated worlds

# H    MINIGRID ENVIRONMENTS FOR OPENAI GYM

The FindObj and MultiRoom environments used for this research are part of MiniGrid, which is an open source gridworld package. This package includes a family reinforcement learning environments compatible with the OpenAI Gym framework. Many of these environments are parameterizable so that the difficulty of tasks can be adjusted (eg: the size of rooms is often adjustable).

## H.1    THE WORLD

In MiniGrid, the world is a grid of size NxN. Each tile in the grid contains exactly zero or one object. The possible object types are wall, door, key, ball, box and goal. Each object has an associated discrete color, which can be one of red, green, blue, purple, yellow and grey. By default, walls are always grey and goal squares are always green.

## H.2    REWARD FUNCTION

Rewards are sparse for all MiniGrid environments. In the MultiRoom environment, episodes are terminated with a positive reward when the agent reaches the green goal square. Otherwise, episodes are terminated with zero reward when a time step limit is reached. In the FindObj environment, the agent receives a positive reward if it reaches the object to be found, otherwise zero reward if the time step limit is reached.

## H.3    ACTION SPACE

There are seven actions in MiniGrid: turn left, turn right, move forward, pick up an object, drop an object, toggle and done. For the purpose of this paper, the pick up, drop and done actions are irrelevant. The agent can use the turn left and turn right action to rotate and face one of 4 possible directions (north, south, east, west). The move forward action makes the agent move from its current tile onto the tile in the direction it is currently facing, provided there is nothing on that tile, or that the tile contains an open door. The agent can open doors if they are right in front of it by using the toggle action.

## H.4    OBSERVATION SPACE

Observations in MiniGrid are partial and egocentric. By default, the agent sees a square of 7x7 tiles in the direction it is facing. These include the tile the agent is standing on. The agent cannot see through walls or closed doors. The observations are provided as a tensor of shape 7x7x3. However, note that these are not RGB images. Each tile is encoded using 3 integer values: one describing the

---

https://github.com/maximecb/gym-minigrid

type of object contained in the cell, one describing its color, and a flag indicating whether doors are open or closed. This compact encoding was chosen for space efficiency and to enable faster training. The fully observable RGB image view of the environments shown in this paper is provided for human viewing.

## H.5 LEVEL GENERATION

The level generation in this task works as follows: (1) Generate the layout of the map (X number of rooms with different sizes (at most size Y) and green goal) (2) Add the agent to the map at a random location in the first room. (3) Add the goal at a random location in the last room. **MultiRoomN$XSY$** - In this task, the agent gets an egocentric view of its surroundings, consisting of $3 \times 3$ pixels. A neural network parameterized as MLP is used to process the visual observation.

**FindObjSY** - In this task, the agent's egocentric observation consists of $7 \times 7$ pixels. We use Convolutional Neural Networks to encode the visual observations.

## I MINIPACMAN ENV

Fuedal Networks primarily has four components: (1) Transition policy gradient, (2) Directional cosine similarity rewards, (3) Goals specified with respect to a learned representation, and (4) dilated RNN. For our work we use normal LSTMs, and hence we do not include design choice (4).

