# OpenReview forum: "InfoBot: Transfer and Exploration via the Information Bottleneck"
_ICLR.cc/2019/Conference_

### Official Review · AnonReviewer1 · 2018-10-18
**Potentially useful but poorly motivated and evaluated**

**Rating:** 3
**Confidence:** 3

**Review:**

The paper proposes a method of regularising goal-conditioned policies with a mutual information term. While this is potentially useful, I found the motivation for the approach and the experimental results insufficient. On top of that the presentation could also use some improvements. I do not recommend acceptance at this time.

The introduction is vague and involves undefined terms such as "useful habits". It is not clear what problems the authors have in mind and why exactly they propose their specific method. The presentation of the method itself is not self-contained and often relies on references to other papers to the point where it is difficult to understand just by reading the paper. Some symbols are not defined, for example what is Z and why is it discrete?

The experiments are rather weak, they are missing comparison to strong exploration baselines and goal-oriented baselines.

---

> ### Author Response · Authors · 2018-11-15
> **Thanks for your feedback!  (1/2)**
>
> Thanks for the feedback. We have conducted additional experiments to address the concerns raised about the evaluation, and we clarify specific points below. We believe that these additions address all of your concerns about the work, though we would appreciate any additional comments or feedback that you might have.
>
> "While this is potentially useful, I found the motivation for the approach  It is not clear what problems the authors have in mind and why exactly they propose their specific method."
>
> We acknowledge that the paper was certainly lacking polish and accept that this may have made the paper difficult to read in places. We have uploaded a revised version in which we have revised the problem statement and writing as per the reviewer's suggestions. We focus on multi-goal environments and goal-conditioned policies. The problem statement is quite simple: we aim to propose an algorithm whereby we incentive agents to learn task structure by training policies that perform well under a variety of goals, while not overfitting to any individual goal. We achieve this by training agents that, in addition to maximizing reward, minimize the policy dependence on the individual goal, quantified by the conditional mutual information  I(A; G | S). In order to minimize this quantity, we formulate it using ideas from variational information bottleneck. To make the paper self-explanatory, we added the mathematical description of the proposed method in the appendix (Section A).
>
> "The presentation of the method itself is not self-contained and often relies on references to other papers to the point where it is difficult to understand just by reading the paper. "
>
> We again acknowledge that the paper was missing certain parts which made the paper difficult to read. We have added another section in the appendix which gives a more mathematical description of the proposed approach. We realized because of the way we have explained things there could be some fundamental misunderstanding about the proposed method. Thus, we would like to clarify this misunderstanding, not only with the intent of convincing you of the idea behind the proposed method but also with the intent of making amends to  the method description where necessary so that readers may not arrive at the same conclusions as you. We added the mathematical description of the proposed framework in the appendix (Section A).

---

> > ### Author Response · Authors · 2018-11-15
> > **More experimental results (Navigation/Comparison to Soft Actor Critic / Multiagent communication)  (2/2)**
> >
> > >> The experiments are rather weak, they are missing comparison to strong exploration baselines and goal-oriented baselines.
> >
> > In order to address reviewer’s concern, we did several  more experiments.
> >
> > We added these experiments to the main paper
> >
> > 1) More challenging Navigation setup.  - We ask the reviewer to refer to heading "More challenging navigation Environment". Also for more details refer to Section 4.6 in the main paper.
> >
> > 2) Comparison to State of the art Off policy Methods (SAC) in sparse  rewards. We ask the reviewer to refer to heading "InfoBot Comparison to State of the art off policy methods (Soft Actor Critic)". Also, for more details refer to  Section F in the appendix.
> >
> > 3) Application of the proposed method in multi-agent communication, such that the goal in the proposed method corresponds to the information obtained because of communication with another agent in multi-agent communication channel. Here,  we  want  to  show  that  by  training  agents  to  develop  “default  behaviours”  as  well  as  the knowledge of when to break those behaviours, using an information bottleneck can also help in other
> > scenarios like multi-agent communication. Consider multiagent communication, where in order to
> > solve a task, agents require communicating with another agents. Ideally, an agent would would like
> > to communicate with other agent, only when its essential to communicate, i.e the agents would like
> > to minimize the communication with another agents. Here we show that selectively deciding when to
> > communicate with another agent can result in faster learning. We follow the same  experimental setup as in the paper (Mordatch and Abbeel, 2018).
> >
> > Method                                                 Train Reward                   TestReward
> > No Communication                               -0.919                                  -0.920
> > Communication                                      -0.36                                   -0.472
> > Communication (with KL cost)            -0.293                                  -0.38
> >
> > (Lower is better). More details about this experimental setup can be found in Section D (Appendix).
> >
> > I. Mordatch and P. Abbeel.  Emergence of grounded compositional language in multi-agent pop-
> > ulations.
> >
> >
> > We would appreciate it if the reviewer could take another look at our changes and additional results, and let us know if the reviewer has request for additional changes that would alleviate the reviewer's concerns.

---

> > > ### Author Response · Authors · 2018-11-21
> > > **InfoBot for Instruction Following - New Result**
> > >
> > > In order to be empirically thorough, we have added another result where we use our method for instruction following i.e the agent has to navigate to a particular goal where the goal is given by the language instruction. We ask the reviewer to refer to heading "InfoBot for Instruction Following" or refer to appendix (section G).
> > >
> > > We would appreciate it if the reviewer could take another look at our changes and additional results, and let us know if the reviewer would like to request additional changes that would alleviate reviewers concerns. We hope that our updates to the manuscript address the reviewer's concerns about clarity, and we hope that the discussion above addresses the reviewer's concerns about empirical significance. We once again thank the reviewer for the feedback of our work.

---

> ### Author Response · Authors · 2018-11-23
> **Feedback by the reviewer. Thanks for your time! :)**
>
> Dear Reviewer,
>
> We appreciate the reviewer's feedback. We would appreciate it if the reviewer could take another look at our changes and additional results, and let us know if the reviewer would like to request additional changes that would alleviate reviewers concerns. We hope that our updates to the manuscript address the reviewer's concerns about clarity, and we hope that the discussion above addresses the reviewer's concerns about empirical significance. We once again thank the reviewer for the feedback of our work.
>
> =====================================================
>
> We have made following changes to the manuscript:
>
> We have updated the paper with the following changes to address reviewer comments:
> - Remove references to "useful habits"
> - Added mathematical description of the proposed method in Appendix (Section A) to answer.
> - Added comparisons to exploration methods (ICM curiosity driven learning) and hierarchical methods (Feudal RL) on a more complicated navigation tasks, which includes branching as well as dead ends.
> - Added comparisons to the state of the art Off policy methods (SAC).
> - Added more experiments showing that the proposed method is more general by using for following scenarios.
>         (1) Using the proposed information regularizer for multi-agent communication, and improving against the strong baseline.
>         (2) Using the proposed method for instruction following where the goal is given by language instruction.
>         (3) Preliminary results showing that the method can be used to provide middle ground b/w model free RL and model based RL.

---

> ### Author Response · Authors · 2018-11-25
> **Request for feedback.**
>
> Your feedback has already been very helpful in improving the paper. Are there any other aspects of the paper that you think could be improved?

---

> ### Author Response · Authors · 2018-12-01
> **Kind request to respond for Reviewer 1**
>
> Dear Reviewer 1,
>
> We thank you again for your informative review that you wrote before the revision period. In our response  we tried our best to address your concerns.  Your feedback has already been very helpful in improving the paper.
>
> The problem statement is quite simple: we aim to propose an algorithm whereby we incentive agents to learn task structure by training policies that perform well under a variety of goals, while not overfitting to any individual goal. We achieve this by training agents that, in addition to maximizing reward, minimize the policy dependence on the individual goal, quantified by the conditional mutual information  I(A; G | S). We show that the proposed method generalizes better by conducting experiments on MANY different problems (Maze based navigation, better regularization for multi-agent communication, better policy for instruction following, middle ground b/w model free and model based methods), as well as we show that adding this specific term  helps the model to identify novel subgoals for further exploration, guiding the agent through a sequence of potential decision  states and through new regions of the state space.
>
> We would highly appreciate to get some feedback from you regarding the changes that we have made and the extra experiments we conducted during the revision period.  In particular, we changed the introduction as you mentioned, and removed references to "useful habits". Then after the revision period you asked the difference b/w the proposed method and DISTRAL (see heading "Difference with Distral (1/2)"), which we believe that we have clarified. And we also clarified the Atari results (see heading, "Clarification regarding Atari Results (2/2)") , and acknowledge that due to the way we have presented these results,  it might have made results a bit difficult to interpret.
>
> Since, the discussion period is coming to end, and the reviewer might also get busy due to NeurIPS, we would highly appreciate a response  and suggestions on how it could be improved. If you still think that paper is uninteresting or not well executed, could you then suggest what specifically it is lacking? Or what result you are looking for essentially ?
>
> We are sincerely hoping to hear from you. We really want to do our best to make sure, we can agree regarding the novelty and clarity of the proposed approach. Thanks very much for your time! :)
>
> The Authors

---

### Official Review · AnonReviewer2 · 2018-11-01
**Interesting idea. Not sure the significance of its experimental results**

**Rating:** 7
**Confidence:** 3

**Review:**

This paper proposes the concept of decision state, which is the state where decision is made “more” dependent to a particular goal. The authors propose a KL divergence regularization to learn the structure of the tasks, and then use this information to encourage the policy to visit the decision states. The method is tested on several different experiment setups.

In general the paper is well-written and easy to follow. Learning a more general policy is not new (as also discussed in the paper), but using the learned structure to further guide the exploration of the policy is novel and interesting.

I have a couple questions about the experimental part though, mostly about the baselines.
1. What is the reasoning behind the selection of the baselines, e.g. A2C as the baseline for the miniGrid experiments?
2. What are the performances of the methods in Table 2, in direct policy generalization? Or is there any reason not reporting them here?
3.  What is the reasoning of picking “Count-base baseline” for Figure 4, rather than the method of curiosity-based exploration?
4. For the Mujoco tasks, there are couple ones outperforming PPO, e.g. TD3, SAC etc.. [1,2] The authors should include their results too.
5. As an ablation study, it would be interesting to see how the bonus reward of visiting decision states can help the exploration on the training tasks, compared to the policy learned from equation (1), and the policies learned without information of other tasks.
6. Lastly, the idea of decision states can also be used in other RL algorithms. It would be also interesting to see if this idea can further improve their performances.

Other comments:
1. Equation (3) should be \le.
2. Why would Equation (5) hold?
3. Right before section 2.2, incomplete sentence.


Disclaimer: The reviewer is not familiar with multitask reinforcement learning, and the miniGrid environment in the paper. Other reviewers should have better judgement on the significance of the experimental results.

[1] Haarnoja, Tuomas, et al. "Soft actor-critic: Off-policy maximum entropy deep reinforcement learning with a stochastic actor." arXiv preprint arXiv:1801.01290 (2018).
[2] Fujimoto, Scott, Herke van Hoof, and Dave Meger. "Addressing Function Approximation Error in Actor-Critic Methods." arXiv preprint arXiv:1802.09477 (2018).

---

> ### Author Response · Authors · 2018-11-15
> **Thanks for your feedback!  (1/3)**
>
> We thank the reviewer for their time and feedback. We  hope to address concerns the reviewer has here.
>
> "What is the reasoning behind the selection of the baselines, e.g. A2C as the baseline for the miniGrid experiments? "
>
> We build from A2C with using goal-conditioned policies and the KL regularization. Hence, The A2C with no kl-regularization is the immediate baseline to consider.  Since for maze experiments, the env is of the  nature of mini-grid POMDP environments with sparse rewards, as well as discrete action, a2c worked out of the box and hence was the most straightforward baseline for comparison.  We would be happy to add other comparisons which reviewer has in mind.
>
>
> >> What are the performances of the methods in Table 2, in direct policy generalization? Or is there any reason not reporting them here?
>
> The setup in direct policy generalization is different as to what we evaluate in Table 2. Direct policy generalization refers to first training an agent with a goal bottleneck on one set of environments (MultiRoomN2S6), and then evaluate the trained agent (without fine tuning on new set of environment ((MultiRoomN3S4, MultiRoomN4S4, and MultiRoomN5S4)).
>
> What we evaluate in Table 2, is how can we transfer the knowledge in form of decision states.
> Basically the intuition is, what we would like from any unsupervised/supervised transferrable exploration technique is to build a policy that is somehow good for adapting to or solving new problems. Here we try to generalize  to new mazes the knowledge acquired by the encoder in the form of the KL estimator. Basically, the intuition is that high KL = interesting state, even before the agent has discovered a single path to the goal. So if we can use egocentric observations and generalize effectively, we can predict which points have high KL before we have even learned to traverse the maze, and then we can use these high-KL regions as rewards without the need to have solved that particular maze in advance, using knowledge transferred from other mazes.
>
> To do this, we first train agents with a goal bottleneck on one set of environments (MultiRoomN2S6) where they learn the sensory cues that correspond to decision states. Then, we use this  knowledge to guide exploration on another set of environments (MultiRoomN3S4, MultiRoomN4S4, and MultiRoomN5S4). And hence in this new environments, we are training another policy from scratch,  And using the KL from D_KL(p(z|s, g) | N(0,1)) as an exploration bonus to guide exploration.

---

> > ### Author Response · Authors · 2018-11-15
> > **Visitation Count  (2/3)**
> >
> > "What is the reasoning of picking “Count-base baseline” for Figure 4, rather than the method of curiosity-based exploration?"
> >
> > The variance across different runs for curiosity based baseline was high, and thats why we reported only count based exploration. Though in table 2, we report the “MAX” performance we got using curiosity driven exploration (over 5 runs), while for InfoBot we average over 5 random seeds.
> >
> > "would be interesting to see how the bonus reward of visiting decision states can help the exploration on the training tasks, compared to the policy learned from equation (1), and the policies learned without information of other tasks."
> >
> > We thank the reviewer for the suggestion. This is indeed an insightful experiment which can help understand the exploration mechanism can effectively identifies decision states, and thus the  model can then identify novel subgoals for further exploration, guiding the agent through a sequence of potential decision  states and through new regions of the state space. Section E in appendix shows that the agent trained with InfoBot visits more diverse states as compared to the baseline agent.

---

> > > ### Author Response · Authors · 2018-11-15
> > > **Decision States (Middle ground b/w Model based and Model free RL) / Comparison to Soft Actor Critic/ Multiagent communication (3/3)**
> > >
> > > The idea of decision states can also be used in other RL algorithms. It would be also interesting to see if this idea can further improve their performances.
> > >
> > > The reviewer is right. The notion of decision points can be used at other places too like planning, combination of model based and model free RL.  Identifying useful decision states can provide a
> > > comfortable middle ground between model-free reasoning and model-based planning. For example,
> > > imagine planning over individual decision states, while using model-free knowledge to navigate
> > > between bottlenecks: aspects of the environment that are physically complex but vary little between
> > > problem instances are handled in a model-free way (the navigation between decision points), while
> > > the particular decision points that are relevant to the task can be handled by explicitly reasoning
> > > about causality, in a more abstract representation. We demonstrate this using a similar setup as in
> > > imagination augmented agents  (Weber et al., 2017). In imagination augmented agents, model free
> > > agents are augmented with imagination, such that the imagination model can be queried to make
> > > predictions about the future. We use the dynamics models to simulate imagined trajectories, which
> > > are then summarized by a neural network and this summary is provided as additional context to
> > > a policy network.  Here, we use the output of the imagination module as a “goal” and we want to
> > > show that only near the decision points (i.e potential subgoals) the agent wants to make use of the
> > > information which is a result of running imagination module).  Here, we want to see, at which points in the state space the policy wants to access the information provided by running the imagination module.  Ideally, only at the decision states (i.e potential sub-goals) policy should access the output of the imagination moduleFor more details, we ask the reviewer to refer to section 4.7 in the main paper.
> > >
> > > (Weber et. al, 2017) - Imagination Augmented Agents  https://arxiv.org/abs/1707.06203
> > >
> > > ======================================================================
> > >
> > > 2) Application of the proposed method in multi-agent communication, such that the goal in the proposed method corresponds to the information obtained because of communication with another agent in multi-agent communication channel. Here,  we  want  to  show  that  by  training  agents  to  develop  “default  behaviours”  as  well  as  the knowledge of when to break those behaviours, using an information bottleneck can also help in other scenarios like multi-agent communication. Consider multiagent communication, where in order to solve a task, agents require communicating with another agents. Ideally, an agent would would like to communicate with other agent, only when its essential to communicate, i.e the agents would like to minimize the communication with another agents. Here we show that selectively deciding when to
> > > communicate with another agent can result in faster learning. We follow the same  experimental setup as in the paper (Mordatch and Abbeel, 2018).
> > >
> > > Method                                                 Train Reward                   TestReward
> > > No Communication                               -0.919                                  -0.920
> > > Communication                                      -0.36                                   -0.472
> > > Communication (with KL cost)            -0.293                                  -0.38
> > >
> > > (Lower is better). More details about this experimental setup can be found in Section D (Appendix).
> > >
> > > I. Mordatch and P. Abbeel.  Emergence of grounded compositional language in multi-agent pop-
> > > ulations.
> > >
> > > ===================================================================================
> > >
> > > 3. Comparison to Soft Actor Critic - We compared the  proposed method  to the state of the art off-policy methods like SAC (Soft Actor Critic). For this domain, we again use high value states as an approximation to the goal state following [1] . In order to implement this, we maintain a buffer of 20000 states, and choose the state with the highest value under the current value function, as a proxy to the goal.
> > >
> > > [1]  Recall Traces, https://arxiv.org/abs/1804.00379
> > > [2]  SAC, Soft Actor Critic https://arxiv.org/abs/1801.01290
> > >
> > > We ask the reviewer to refer to heading "InfoBot Comparison to State of the art off policy methods (Soft Actor Critic)". Also, for more details refer to  Section F in the appendix.
> > >
> > > Closing:
> > > Thank you for your time. We hope you find that our revision addresses your concerns.
> > >  We hope that our updates to the manuscript address the reviewer's concerns about clarity, and we hope that the discussion above addresses the reviewer's concerns about empirical significance. We once again thank the reviewer for the thorough feedback of our manuscript. Please let us know if anything is unclear here, if you’re uncertain about part of the argument, or if there is any other comparison that would be helpful in clarifying things more.

---

> > > > ### Comment · AnonReviewer2 · 2018-11-19
> > > > **Thanks for the rebuttal**
> > > >
> > > > I have read other reviews and I think the rebuttal has addressed my concerns on the significance of the experimental results, thus I increase the score from 6 to 7.

---

> > > > > ### Author Response · Authors · 2018-11-20
> > > > > **Thanks!**
> > > > >
> > > > > We thank the reviewer for taking time to read our lengthy rebuttal, and increasing their score. If the reviewer want us to include any more ablation/experiment, please let us know. Thanks again! :)

---

> ### Author Response · Authors · 2018-11-19
> **InfoBot for Instruction Following - New Result**
>
> We have added another result where we use our method for instruction following i.e the agent has to navigate to a particular goal where the goal is given by the language instruction. We ask the reviewer to refer to heading "InfoBot for Instruction Following" or refer to appendix (section G).
>
> We would appreciate it if the reviewer could take another look at our changes and additional results, and let us know if the reviewer would like to request additional changes that would alleviate reviewers concerns. We hope that our updates to the manuscript address the reviewer's concerns about clarity, and we hope that the discussion above addresses the reviewer's concerns about empirical significance. We once again thank the reviewer for the thorough feedback of our work.

---

> ### Author Response · Authors · 2018-11-24
> **Feedback by reviewer. Thanks for your time! :)**
>
> Dear Reviewer,
>
> We appreciate the reviewer's feedback. We would appreciate it if the reviewer could take another look at our changes and additional results, and let us know if the reviewer would like to request additional changes that would alleviate reviewers concerns. We hope that our updates to the manuscript address the reviewer's concerns about clarity, and we hope that the discussion above addresses the reviewer's concerns about empirical significance. We once again thank the reviewer for the feedback of our work.
>
> ===============================================================================================
>
> We have made following changes to the manuscript:
>
> We have updated the paper with the following changes to address reviewer comments:
> - Added mathematical description of the proposed method in Appendix (Section A) to answer.
> - Added comparisons to exploration methods (ICM curiosity driven learning) and hierarchical methods (Feudal RL) on a more complicated navigation tasks, which includes branching as well as dead ends.
> - Added comparisons to the state of the art Off policy methods (SAC).
> - Added more experiments showing that the proposed method is more general by using for following scenarios.
>         (1) Using the proposed information regularizer for multi-agent communication, and improving against the strong baseline.
>         (2) Using the proposed method for instruction following where the goal is given by language instruction.
>         (3) Preliminary results showing that the method can be used to provide middle ground b/w model free RL and model based RL.

---

### Official Review · AnonReviewer3 · 2018-11-14
**Review Number 3 (So sorry for the delay!)**

**Rating:** 7
**Confidence:** 3

**Review:**

The authors propose a new regularizer for policy search in a multi-goal RL setting. The objective promotes a more efficient exploration strategy by encouraging the agent to learn policies that depend as little as possible on the target goal. This is achieved by regularizing standard RL losses with the negative conditional mutual information I(A;G|S). Although this regularizer cannot be optimize, the authors propose a tractable bound. The net effect of this regularizer is to promote more effective exploration by encouraging the agent to visit decision states, in which goal-depend decisions play a more important role. The idea of using this particular regularizer is inspired by an existing line of work on the information bottleneck.

I find the idea proposed by the authors to be interesting. However, I have the following concerns, and overall I think this paper is borderline.

1. The quality of the experimental validation provided by the authors is in my opinion borderline acceptable. Although the method performs better on toy settings, it seems barely better on more challenging ones. Experiments in section 4.5 lack detail and context.
2. The clarity of the presentation is also not great.
    2.1. The two-stage nature of the method was confusing to me. I didn’t understand the role of the second stage. Most focus is on the first stage, and only very little on the second stage. For example, I was confused about why the sign of the regularizer was flipped.
    2.2. I was confused by how exactly the bounds (3) and (4) we applied and in what order.
    2.3. I think the intuition of the method could be better explained and better validated by experiments.

I also have the following additional comments:
* How is the regularizer applied with other policy search algorithms besides Reinforce? Was it done in the paper? I can’t say for sure. Specifically, when comparing to PPO, was the algorithm compared to a version of PPO augmented with this regularizer? Why yes or why no?
* More generally, experiments where more modern policy search algorithms are combined with the regularizer would be helpful. In particular, does it matter which policy search algorithm we use with this method?
* Experimental plots in section 4.4 are missing error bars, and I can’t tell if the results are significant without them.
* I thought the motivation for choosing this regularizer was lacking. The authors cite the information bottleneck literature, but we shouldn’t need to read all these papers, the main ideas should be summarized here.
* The argument for how the regularizer improves exploration seemed to me very hand-wavy and not well substantiated by experiments.
* I would love to see a better discussion of how the method is useful when he RL setting is not truly multi-goal.
* The second part of the algorithm needs to be explained much more clearly.
* What is the effect of the approximation on Q?

---

I have read the response of the authors, and they have addressed a significant numbers of concerns that I had. I am upgrading my rating to a 7.

---

> ### Author Response · Authors · 2018-11-15
> **Thanks for your feedback!  (1/4)**
>
> We thank the reviewer for the positive and constructive feedback. We appreciate that the reviewer finds that our method interesting.
>
> "The quality of the experimental validation provided by the authors is in my opinion borderline acceptable. Although the method performs better on toy settings, it seems barely better on more challenging ones. Experiments in section 4.5 lack detail and context."
>
> In order to make our experiments more rigorous, we conducted more experiments to answer reviewers concern.
>
> 1) More challenging Navigation setup.  - We ask the reviewer to refer to heading "More challenging navigation Environment". Also for more details refer to Section 4.6 in the main paper.  We compared to several strong baselines like hierarchical RL methods, strong exploration methods as well as goal based methods. We would be happy to add more comparisons which reviewer has in mind.
>
> 2) Comparison to State of the art Off policy Methods (SAC) in sparse  rewards. We ask the reviewer to refer to heading "InfoBot Comparison to State of the art off policy methods (Soft Actor Critic)". Also, for more details refer to  Section F in the appendix.
>
> 3) Application of the proposed method in multi-agent communication, such that the goal in the proposed method corresponds to the information obtained because of communication with another agent in multi-agent communication channel. Here,  we  want  to  show  that  by  training  agents  to  develop  “default  behaviours”  as  well  as  the knowledge of when to break those behaviours, using an information bottleneck can also help in other
> scenarios like multi-agent communication. Consider multiagent communication, where in order to
> solve a task, agents require communicating with another agents. Ideally, an agent would would like
> to communicate with other agent, only when its essential to communicate, i.e the agents would like
> to minimize the communication with another agents. Here we show that selectively deciding when to
> communicate with another agent can result in faster learning. We follow the same  experimental setup as in the paper (Mordatch and Abbeel, 2018).
>
> Method                                                 Train Reward                   TestReward
> No Communication                               -0.919                                  -0.920
> Communication                                      -0.36                                   -0.472
> Communication (with KL cost)            -0.293                                  -0.38
>
> (Lower is better). More details about this experimental setup can be found in Section D (Appendix).
>
> I. Mordatch and P. Abbeel.  Emergence of grounded compositional language in multi-agent pop-
> ulations.
>
> We acknowledge that the paper was certainly lacking polish and accept that this may have made the paper difficult to read in places. We have uploaded a revised version in which we have revised the problem statement and writing as per the reviewer's suggestions.

---

> > ### Author Response · Authors · 2018-11-15
> > **More Intuition (2/4)**
> >
> > "I thought the motivation for choosing this regularizer was lacking. The authors cite the information bottleneck literature, but we shouldn’t need to read all these papers, the main ideas should be summarized here."
> >
> > We focus on multi-goal environments and goal-conditioned policies. The problem statement is quite simple: we aim to propose an algorithm whereby we incentive agents to learn task structure by training policies that perform well under a variety of goals, while not overfitting to any individual goal. We achieve this by training agents that, in addition to maximizing reward, minimize the policy dependence on the individual goal, quantified by the conditional mutual information  I(A; G | S). In order to minimize this quantity, we formulate it using ideas from variational information bottleneck. To make the paper self-explanatory, we added the mathematical description of the proposed method in the appendix (Section A).
> >
> > "The argument for how the regularizer improves exploration seemed to me very hand-wavy and not well substantiated by experiments."
> >
> >
> > We evaluate in Table 2 how the regularizer improves exploration, the basic idea is how can we transfer the knowledge in form of decision states. Basically the intuition is, what we would like from any unsupervised/supervised transferrable exploration technique is to build a policy that is somehow good for adapting to or solving new problems. Here we try to generalize  to new mazes the knowledge acquired by the encoder in the form of the KL estimator. Basically, the intuition is that high KL = interesting state, even before the agent has discovered a single path to the goal. So if we can use egocentric observations and generalize effectively, we can predict which points have high KL before we have even learned to traverse the maze, and then we can use these high-KL regions as rewards without the need to have solved that particular maze in advance, using knowledge transferred from other mazes.
> >
> > To do this, we first train agents with a goal bottleneck on one set of environments (MultiRoomN2S6) where they learn the sensory cues that correspond to decision states. Then, we use this  knowledge to guide exploration on another set of environments (MultiRoomN3S4, MultiRoomN4S4, and MultiRoomN5S4). And hence in this new environments, we are training another policy from scratch,  And using the KL from D_KL(p(z|s, g) | N(0,1)) as an exploration bonus to guide exploration. As shown in Table 2, We outperform strong exploration methods like Curiosity Driven Learning (ICM) and count based exploration. We would be happy to add other comparisons which reviewer has in mind.
> >
> > We did another experiment in which we  show how the bonus reward of visiting decision states can help the exploration on the training tasks as compared to the policy learned from equation (1), and the policies learned without information of other tasks. The intuition is that  once the agent has been trained using goal bottleneck, then in the transfer state, the proposed  model can identify novel subgoals for further exploration using the learned knowledge of decision states in part 1, thus guiding the agent through a sequence of potential decision  states and through new regions of the state space.  We show that the agent trained with InfoBot visits more diverse states as compared to the baseline agent. We ask the reviewer to refer to Section E in appendix.
> >
> > "How is the regularizer applied with other policy search algorithms besides Reinforce? Was it done in the paper? I can’t say for sure. Specifically, when comparing to PPO, was the algorithm compared to a version of PPO augmented with this regularizer? Why yes or why no?"
> >
> > Yes, the proposed method can be combined with any policy based method (such that we have a parameterized form for the policy). In our experiments, we combine this with a2c for Minigrid Env as well as Atari setup, PPO for mujoco based envs, as well as state of the art off policy algorithms like soft actor critic. (SAC). Since, our method relies on goals, in order to minimize the I(A;G|S), in Atari as well as mujoco domains, we use high value states as a proxy to the goal state following [1] . In order to implement this, we maintain a buffer of 20000 high value states, and choose the state with the highest value under the current value function, as a proxy to the goal.
> >
> > [1]  Recall Traces, https://arxiv.org/abs/1804.00379

---

> > > ### Author Response · Authors · 2018-11-15
> > > **Proposed Method is general and can be combined with a2c/ppo/TD3/SAC. (3/4)**
> > >
> > > As shown in the experiments, our method outperforms the baseline in the Atari case, as well as compared to PPO for continuous control tasks (Section 4.4), as well as Soft Actor Critic on Sparse reward Continuous control problems.
> > >
> > > "More generally, experiments where more modern policy search algorithms are combined with the regularizer would be helpful. In particular, does it matter which policy search algorithm we use with this method?"
> > >
> > > We combined the proposed method with state of the art off policy algorithm (Soft actor Critic), PPO and A2C. In general, the proposed method is agnostic to the chosen policy search algorithm. We would be happy to add more comparisons, which the reviewer has in mind.
> > >
> > > "Experimental plots in section 4.4 are missing error bars, and I can’t tell if the results are significant without them."
> > >
> > > Experimental results in section 4.4 are averaged over 5 random seeds. We found the baseline (PPO) as well as proposed method to have less variance across different runs with different random seeds. Since, we were plotting many baselines (InfoBot with Low value states, InfoBot with High Value States, InfoBot with zero KL cost), and hence for clarity, we did not include the error bars. But if the reviewer wants, we would be happy to include them.
> > >
> > > We would appreciate it if the reviewer could take another look at our changes and additional results, and let us know if the reviewer would like to request additional changes that would alleviate reviewers concerns. We hope that our updates to the manuscript address the reviewer's concerns about clarity, and we hope that the discussion above addresses the reviewer's concerns about empirical significance. We once again thank the reviewer for the thorough feedback of our work.

---

> > > > ### Author Response · Authors · 2018-11-19
> > > > **InfoBot for Instruction Following - New Result (4/4)**
> > > >
> > > > We have added another result where we use our method for instruction following i.e the agent has to navigate to a particular goal where the goal is given by the language instruction. We ask the reviewer to refer to heading "InfoBot for Instruction Following" or refer to appendix (section G).
> > > >
> > > > We would appreciate it if the reviewer could take another look at our changes and additional results, and let us know if the reviewer would like to request additional changes that would alleviate reviewers concerns. We hope that our updates to the manuscript address the reviewer's concerns about clarity, and we hope that the discussion above addresses the reviewer's concerns about empirical significance. We once again thank the reviewer for the feedback of our work.

---

> ### Author Response · Authors · 2018-11-23
> **Feedback by reviewer. Thanks for your time! :)**
>
> Dear Reviewer,
>
> We appreciate the reviewer's feedback. We would appreciate it if the reviewer could take another look at our changes and additional results, and let us know if the reviewer would like to request additional changes that would alleviate reviewers concerns. We hope that our updates to the manuscript address the reviewer's concerns about clarity, and we hope that the discussion above addresses the reviewer's concerns about empirical significance. We once again thank the reviewer for the feedback of our work.
>
> ===============================================================================================
>
> We have made following changes to the manuscript:
>
> We have updated the paper with the following changes to address reviewer comments:
> - Added mathematical description of the proposed method in Appendix (Section A) to answer.
> - Added comparisons to exploration methods (ICM curiosity driven learning) and hierarchical methods (Feudal RL) on a more complicated navigation tasks, which includes branching as well as dead ends.
> - Added comparisons to the state of the art Off policy methods (SAC).
> - Added more experiments showing that the proposed method is more general by using for following scenarios.
>         (1) Using the proposed information regularizer for multi-agent communication, and improving against the strong baseline.
>         (2) Using the proposed method for instruction following where the goal is given by language instruction.
>         (3) Preliminary results showing that the method can be used to provide middle ground b/w model free RL and model based RL.

---

> ### Author Response · Authors · 2018-12-01
> **Kind request to respond for Reviewer 3**
>
> Dear Reviewer 3,
>
> We thank you again for your informative review that you wrote before the revision period. In our response  we tried our best to address your concerns.  Your feedback has already been very helpful in improving the paper.
>
> The problem statement is quite simple: we aim to propose an algorithm whereby we incentive agents to learn task structure by training policies that perform well under a variety of goals, while not overfitting to any individual goal. We achieve this by training agents that, in addition to maximizing reward, minimize the policy dependence on the individual goal, quantified by the conditional mutual information  I(A; G | S). We show that the proposed method generalizes better by conducting experiments on MANY different problems (Maze based navigation, better regularization for multi-agent communication, better policy for instruction following, middle ground b/w model free and model based methods), as well as we show that adding this specific term  helps the model to identify novel subgoals for further exploration, guiding the agent through a sequence of potential decision  states and through new regions of the state space.
>
> Since, the discussion period is coming to end, and the reviewer might also get busy due to NeurIPS, we would highly appreciate a response  and suggestions on how it could be improved. If you still think that paper is not well executed, could you then suggest what specifically it is lacking? Or what result you are looking for essentially ?
>
> We are sincerely hoping to hear from you.  Thanks very much for your time! :)
>
> The Authors

---

> ### Author Response · Authors · 2018-12-17
> **Thanks!**
>
> The authors thank the reviewer for reading the rebuttal, and increasing their score.
>
> Thanks for your time!

---

### Author Response · Authors · 2018-11-15
**More challenging navigation Environment - To ALL reviewers**

We thank all the reviewers for their valuable time and feedback.

In order to answer the  questions raised by all the reviewers, we setup another challenging task (Fig 7 main paper). Here, the goal of the agent is to navigate to a goal position. We use a partially observed formulation of the task, where the agent only observes a small number of squares ahead of it and the agent only gets a reward of “1” when it reaches the goal.  For standard RL algorithms, these tasks are difficult to solve due to the partial observability of the environment, sparse reward (as the agent receives a reward only after reaching the goal), and low probability of reaching the goal via random walks (precisely because these  junction states are crucial states where the right action must be taken and several junctions need to be crossed). This task also has dead ends as well as more complex branching factor which was not present in the task which we tried for the submitted paper.  We demonstrate the results on this env where we compare generalization performance to model free methods (a2c, ppo), and goal oriented baselines (UVFA) setup, as well as strong exploration baselines (ICM, count based exploration) and goal based hierarchical baselines (like Option Critic and Feudal Networks). We would be happy to add comparisons to other baselines which reviewers have in mind.

We first show direct policy transfer in which we first train a particular algorithm on 6 x 6 maze, and evaluate the direct policy transfer on 11 x 11 maze. In order to be exhaustive, we also compare to hierarchical baselines like Option Critic Architecture and Feudal Networks. We tested on 11 X 11 mazes by running a policy trained with different algorithms. The success rate is the number of times the agent solves a larger task (11 x 11) while it is being trained on the smaller task (6 x 6).  All these were run for 100M steps, and averaged over 3 random seeds.

Algorithm (Train on 6 x 6)                                                    (Evaluate on 11 x 11) (variance)
Actor Critic                                                                                               5% (1%)
PPO (Proximal Policy Optimization)                                                    8% (1%)
Actor Critic + CT based exploration                                                     7% (1%)
Goal Based (UVFA)                                                                                15% (4%)
Curiosity Driven Learning using inverse models.                            47% (5%)
Option Critic                                                                                            17% (4%)
Feudal RL                                                                                                37% (4%)
Proposed Method                                                                                 64% (2%)

When generalizing to larger mazes, the agent learns to solve the task  64\% of the times, whereas other hierarchical agents solve <50\% of mazes (Table 3).

---

> ### Author Response · Authors · 2018-11-15
> **InfoBot Comparison to State of the art off policy methods (Soft Actor Critic) - To ALL reviewers**
>
> We also compared the  proposed method  to the state of the art off-policy methods like SAC. For this domain, we again use high value states as an approximation to the goal state following [1] . In order to implement this, we maintain a buffer of 20000 states, and choose the state with the highest value under the current value function, as a proxy to the goal.
>
> We compare the proposed method with SAC in  sparse reward scenarios. We evaluate the proposed algorithm  on 4 MuJoCo tasks in OpenAI Gym.
>
> The result in Figure. 11 (Section F appendix) shows the performance of the proposed method showing improvement over the baseline on the sparse reward HalfCheetah-v2, Walker2d-v2, Swimmer-v2, Hopper-v2 tasks. We did not observe much benefit over SAC in dense reward scenario. Unlike Minigrid, and maze goal based navigation task, the reward structure in this benchmark is  dense in that the agent always receives a reasonable amount of reward according to its continuous progress. Hence, in the dense reward scenario, the learning problem for the policy regularized with goal bottleneck is not simpler as compared to that of the regular policy, and hence the  agent may not be forced to generalize across different contexts (by learning ``useful'' habits).
>
> To verify our conjecture, we conducted experiments by making the reward which agent gets from the environment sparse. More specifically, the modified tasks give an reward once in 50 steps. We see that in this scenario, the agent trained with goal bottleneck regularization performs much better as compared to the SAC baseline.
>
>
> [1]  Recall Traces, https://arxiv.org/abs/1804.00379
> [2]  SAC, Soft Actor Critic https://arxiv.org/abs/1801.01290

---

> > ### Author Response · Authors · 2018-11-19
> > **InfoBot for Instruction Following - To ALL REVIEWERS**
> >
> > Here we want to show that the proposed method can also be in the context of interactive worlds for spatial reasoning where the goal is given by language instruction, and hence the proposed method is very general. The agent is placed in an interactive world, and agent can take actions to reach the goal specified by language instruction. For ex. Reach the north-most house, the problem could be challenging because the language instruction is is highly context dependent. Therefore, for better generalization to unseen worlds, the model must jointly reason over the instruction text and environment configuration. Here, the text instructions can have both local and global references to objects. Local references require an understanding of spatial prepositional phrases such as ‘above’, ‘next to’ in order to reach the goal. This is invariant to the global position of the object references, on the other hand, global references contains superlatives such as ‘easternmost’ and ‘topmost’, which require reasoning over the entire map. Here we compare the proposed method to UVFA. We use the exact same experimental setup as in [1].
> >
> > Method                                          Local (Policy Quality)                         Global (Policy Quality)
> > UVFA                                                         0.57                                                     0.59
> > InfoBot                                                     0.89                                                     0.81
> >
> > [1] "Representation Learning for Grounded Spatial Reasoning" https://arxiv.org/abs/1707.03938
> >
> > The details for all these experiments have also been mentioned in Appendix (Section G).
> >
> > We would appreciate it if the reviewer(s) could take another look at our changes and additional results, and let us know if the reviewer would like to request additional changes that would alleviate reviewers concerns.

---

### Author Response · Authors · 2018-11-19
**Paper Updated to address reviewer feedback.**

We have updated the paper with the following changes to address reviewer comments:
- Remove references to "useful habits"  (Reviewer 1)
- Added mathematical description of the proposed method in Appendix (Section A) to answer (Reviewer 1, and Reviewer 3)
- Added comparisons to exploration methods (ICM curiosity driven learning) and hierarchical methods (Feudal RL) on a more complicated navigation tasks, which includes branching as well as dead ends. (All reviewers)
- Added comparisons to the state of the art Off policy methods (SAC).  (All reviewers)
- Added more experiments showing that the proposed method is more general by using for following scenarios. (All Reviewers)
        (1) Using the proposed information regularizer for multi-agent communication, and improving against the strong baseline.
        (2) Using the proposed method for instruction following where the goal is given by language instruction.
        (3) Preliminary results showing that the method can be used to provide middle ground b/w model free RL and model based RL.

Thank you for your time! The authors appreciate the time reviewers have taken for providing feedback. which resulted in improving the presentation of our paper. Hence,  we would appreciate it if the reviewers could take a look at our changes and additional results, and let us know if they would like to either revise their rating of the paper, or request additional changes that would alleviate their concerns.

---

### Author Response · Authors · 2018-12-02
**Final Rebuttal Summary**

Since today is the last day for the discussion period,  we want to summarize our rebuttal.

First, We want to thank all reviewers for their critical feedback and suggestions, which has already helped us improve the paper’s clarity and presentation. Both the reviewers (R1 and R2) agree that the paper tackles an interesting  problem. The main concerns were about the clarity of the writing (R1 and R3) , making it hard to clearly assess the underlying contributions, and about the diversity of the experimental results. (all the reviewers).

We aim to propose an algorithm whereby we incentive agents to learn task structure by training policies that perform well under a variety of goals, while not overfitting to any individual goal. We achieve this by training agents that, in addition to maximizing reward, minimize the policy dependence on the individual goal, quantified by the conditional mutual information  I(A; G | S).  Thus, we claim that adding this mutual information acts as a regularizer which promotes generalization across tasks, and it helps in achieve better exploration.

We show that the proposed method generalizes better by conducting experiments on MANY different problems during the rebuttal time (Maze based navigation, better regularization for multi-agent communication, better policy for instruction following, middle ground b/w model free and model based methods, comparisons to state of the art off policy methods), as well as we show that adding this specific term  helps the model to identify novel subgoals for further exploration, guiding the agent through a sequence of potential decision  states and through new regions of the state space. Thus, we show experimental results to back up our claims in our submission.

============================

-  The key point in R3 and R2 review was lack of experimental results. We believe that we did conducted many more experiments to show that adding I(A; G|S) helps in better generalization as well as exploration. And we are also thankful to R2 as they increased their score.

- The key point in R1's review was lack of clarity (before the revision period) which we believe we have tried our best to address. And then after the revision period, the reviewer asked comparison to DISTRAL and clarity about Atari results, which we again believe we have addressed.

We urge R1 and R3  to read our rebuttal as we have carefully addressed and incorporated all critical feedback and suggestions.

Thanks very much for your time, and feedback which has improved the presentation of our paper! :-)

---

### Meta-Review · Area_Chair1 · 2018-12-15
**An interesting link between generalization and exploration**

**Confidence:** 3
**Recommendation:** Accept (Poster)

**Metareview:**

The paper presents the use of information bottlenecks as a way to identify key "decision states" in exploration, in a goal-conditioned model. The concept of "decision states" is actually common in RL, states where exploring can lead to very diverse/new states. The implementation of the "information bottleneck" is done by adding a regularizing term, the conditional mutual information I(A;G|S).

The main weaknesses of the paper were its lack of clarity and the experimental section. It seems to me that the rebuttals, and the additional experiments and details, made the paper worthy of publication. The authors cleared enough of the gray areas and showcased the relative merits of the methods.